# Determining Genetic Diversity and Population Structure of Common Bean (*Phaseolus vulgaris* L.) Landraces from Türkiye Using SSR Markers

**DOI:** 10.3390/genes13081410

**Published:** 2022-08-08

**Authors:** Güller Özkan, Kamil Haliloğlu, Aras Türkoğlu, Halil Ibrahim Özturk, Erdal Elkoca, Peter Poczai

**Affiliations:** 1Department of Biology, Faculty of Science, Ankara University, Ankara 06100, Türkiye; 2Department of Field Crops, Faculty of Agriculture, Ataturk University, Erzurum 25240, Türkiye; 3Department of Biology, Faculty of Science, Cankiri Karatekin University, Çankırı 18200, Türkiye; 4Department of Field Crops, Faculty of Agriculture, Necmettin Erbakan University, Konya 42310, Türkiye; 5Health Services Vocational School, Binali Yıldırım University, Erzincan 24100, Türkiye; 6Vocational High School, Department of Plant and Animal Production, İbrahim Çeçen University, Ağrı 04100, Türkiye; 7Botany Unit, Finnish Museum of Natural History, University of Helsinki, FI-00014 Helsinki, Finland; 8Institute of Advanced Studies Kőszeg (iASK), H-9731 Kőszeg, Hungary

**Keywords:** bean breeding, genetic diversity, molecular markers, structure

## Abstract

Assessment of genetic diversity among different varieties helps to improve desired characteristics of crops, including disease resistance, early maturity, high yield, and resistance to drought. Molecular markers are one of the most effective tools for discovering genetic diversity that can increase reproductive efficiency. Simple sequence repeats (SSRs), which are codominant markers, are preferred for the determination of genetic diversity because they are highly polymorphic, multi-allelic, highly reproducible, and have good genome coverage. This study aimed to determine the genetic diversity of 40 common bean (*Phaseolus vulgaris* L.) landraces collected from the Ispir district located in the Northeast Anatolia region of Türkiye and five commercial varieties using SSR markers. The Twenty-seven SSR markers produced a total of 142 polymorphic bands, ranging from 2 (GATS91 and PVTT001) to 12 (BM153) alleles per marker, with an average number of 5.26 alleles. The gene diversity per marker varied between 0.37 and 0.87 for BM053 and BM153 markers, respectively. When heterozygous individuals are calculated proportional to the population, the heterozygosity ranged from 0.00 to 1.00, with an average of 0.30. The expected heterozygosity of the SSR locus ranged from 0.37 (BM053) to 0.88 (BM153), with an average of 0.69. Nei’s gene diversity scored an average of 0.69. The polymorphic information content (PIC) values of SSR markers varied from 0.33 (BM053) to 0.86 (BM153), with an average of 0.63 per locus. The greatest genetic distance (0.83) was between lines 49, 50, 53, and cultivar Karacaşehir-90, while the shortest (0.08) was between lines 6 and 26. In cluster analysis using Nei’s genetic distance, 45 common bean genotypes were divided into three groups and very little relationship was found between the genotypes and the geographical distances. In genetic structure analysis, three subgroups were formed, including local landraces and commercial varieties. The result confirmed that the rich diversity existing in Ispir bean landraces could be used as a genetic resource in designing breeding programs and may also contribute to Türkiye bean breeding programs.

## 1. Introduction

Bean (*Phaseolus vulgaris* L.) is one of the most important cultivated plants from the legume family worldwide in terms of total yield and cultivated area [1]. Beans consumed in different forms (green pods, immature or dried seeds) are a primary source of vegetable protein in the human diet [2]. For many people living in European countries, *P. vulgaris* is a traditional dietary component [3,4].

Bean has two centers of genetic diversity that are distinguished and recognized according to their phenotypic, biochemical, and genotypic differences. These are Middle American and Andean gene centers. These centers of genetic diversity are separated from each other by both geographical and partial reproductive barriers [1,5]. In addition, these gene pools vary strongly with morphological and biochemical markers [6]. In terms of genetic diversity, the Mesoamerican gene pool has more diversity than the Andean gene pool [7]. Seeds of beans in the Middle American gene pool are characterized as either small or medium-sized, while seeds of phenotypes in the Andean gene pool are usually described as larger [8]. It is thought that the common bean was introduced to Europe in the 16th and 17th centuries, and it was introduced to Türkiye in the 17th century [9].

People living in different regions of Türkiye have been cultivating beans for centuries, and it is an important food in their diet. Local bean genotypes are still used by people living in many rural areas of the country. Beans have spread to many regions of the country with both natural and artificial selections over time and have created populations known with names specific to these regions [2]. Beans show marked genetic variation in terms of many morphological characteristics such as seed, pod, and flower characteristics [10]. However, the increase in the use of commercial varieties in recent years and the insufficient use of local varieties in breeding programs have led to a significant narrowing of the genetic base [11,12]. Therefore, knowledge of genetic diversity, examination of population structure, and understanding of the relationships of varieties within and between commercial classes are essential steps for genetic improvement and preservation of the genetic diversity of beans [13]. In addition to being the center of domestication for important Old-World cereal and grain legume crops with its ecological and geographical characteristics, Türkiye has an important place globally regarding plant genetic diversity. Beans, which have a critical role in Türkiye in both the economy and human nutrition, have a high genetic diversity [14].

Genetic diversity studies are important for breeding programs since they provide valuable information for the effective conservation and application of existing germplasms [15]. Such studies facilitate understanding of genetic relationships between accessions, identification of germplasm excesses and admixtures, and identification of genitor pairs with sufficient genetic distance [16]. Evaluation of morphological traits is a traditional method to identify and define the relationship between local bean genotypes. Genotypes can be divided into groups according to their morphological characteristics, e.g., fruit flesh, flower structure, leaf shape, and seed [2]. However, morphological features may change under the influence of ecological conditions. Therefore, the use of these markers is not fully effective in determining diversity. In recent years, molecular markers have been used together with morphological markers to evaluate genetic diversity in plants. Because molecular markers are not affected by ecological factors, it is an important method for genetic mapping and examining genetic diversity, population structure, and phylogenetic relationships in many plants such as beans [17,18].

Detection of genetic diversity and higher genetic diversity provide extremely important information in selecting superior genotypes for plant breeding [19]. Many molecular methods, such as random amplified polymorphic DNA (RAPD) [20], inter-simple sequence repeat (ISSR) [21], amplified fragment length polymorphism (AFLP) [22], simple-sequence repeats (SSR) [4], inter-primer binding site (iPBS) retrotransposon [23], single-nucleotide polymorphisms (SNPs) [12], and start codon targeted markers (SCoT) [24], are used for the analysis of genetic diversity and population structure in beans. Among molecular markers, SSRs have special relevance in analyzing genetic diversity [25]. SSRs, highly polymorphic and codominant markers [26], have short repeating DNA sequences, usually 2–6 bp in length, and are used in genome mapping, gene tagging, genetic diversity estimation, variety identification, and marker-assisted selection [27]. SSRs are widely and successfully used to determine genetic diversity in beans and to create genetic maps [4,13,28,29,30]. in reviewed recent studies, the bean genotypes evaluated by SSR markers used only a narrow or different region source. In addition, a wide-ranging study has not yet been conducted to measure the genetic diversity of bean germplasm in Türkiye. This research was carried out to reveal the genetic diversity and population structure of the landraces obtained from the bean population grown in the Ispir district by using the SSR molecular marker method to determine the degree of inbreeding between the landraces and to reveal suitable lines for breeding studies. Therefore, the germplasm information yielded from this study will be useful for bean breeding studies. In addition, the findings are anticipated to contribute to the development of strategies to protect endangered bean genetic resources in the Erzurum-Ispir region.

## 2. Materials and Methods

### 2.1. Plant Material

The common bean genotypes used in this study were collected from the Ispir Valley in Northeast Anatolia, Türkiye. A total of 45 common bean genotypes together with five nationally registered cultivars were used for SSR analysis. (Figure 1 and Table 1). In addition, some seed characteristics (100-seed weight, seed color, and seed shape) of bean accessions are presented in Table 1.

### 2.2. DNA Extraction

Sample plants were grown in a greenhouse of the Atatürk University Field Crops Department. Bulk DNA of 45 individuals per accession was prepared from young leaves of 2-week-old plants in the Laboratory of Molecular Biology and Genetics, Department of Field Crops, Ataturk University, Türkiye. Genomic DNA extractions were performed as described by Zeinalzadehtabrizi et al. [31]. DNA quality was affirmed through electrophoresis in 0.8% agarose gel. The NanoDrop^®^ ND-1000 UV/V spectrophotometer (Thermo Fisher Scientific, Waltham, MA, USA) was used to determine DNA concentrations. For SSR analysis, the final DNA concentration was adjusted to 50 µg/mL. Diluted DNA samples were stored at −20 °C to await SSR-polymerase chain reaction (PCR).

### 2.3. SSR Analysis

Twenty-seven SSR primer pairs were selected from previous studies based on their reliable amplification patterns and high polymorphic information contents. There are three markers on the Pv01 chromosome, five markers on the Pv02 chromosome, five markers on the Pv04 chromosome, two markers on the Pv06 chromosome, and two markers on the Pv07 chromosome, three markers on the Pv08 chromosome, and five markers on the Pv09 chromosome. There was only one marker on the Pv01 (BM053 marker) and Pv05 (BM175 marker) chromosomes. In addition, none of the markers used in our study were markers located on the Pv10 and Pv11 chromosomes (Table 2). These primer pairs resulted in specific and stable DNA profiles in this study. PCR amplifications were performed in Labcycler. The PCR mixture consisted of 10× buffer, 2 mM MgCl_2_, 0.25 mM of each dNTP, 2 µM (20 pmol) primer, 0.5 U Taq polymerase, and 50 µg/ng DNA template in a 20 µL reaction mixture. The amplification conditions were as follows: an initial denaturation step of 2 min at 95 °C, 37 cycles of 30 s at 95 °C, 60 s at 47–58 °C and 60 s at 72 °C, and a final extension step of 5 min at 72 °C. The amplification products were resolved on 3% agarose gel in 1X SB buffer at 150 V/cm for 120 min, stained with ethidium bromide (0.2 ug/mL), visualized under a UV-transilluminator, and photographed under ultraviolet light with Nikon Coolpix5000. The sizes of the base pairs were determined based on a DNA ladder between 50 and 1000 bp (Vivantis product no. NM2421) [32].

### 2.4. Molecular Data Analysis

Scoring was given as 1 (presence) and 0 (absence) for amplified fragments at each SSR locus, and data matrices were constructed accordingly. In this study, Phylogenetic analysis was performed with MEGA software (v. 7.0.14). In this study, Phylogenetic analysis was performed with MEGA 6.0 software. The dendrogram was constructed using the neighbor-joining method of the MEGA software with the maximum composite likelihood substitution model, and bootstrapping with 1000 replicates [38]. Marker index for SSR markers was calculated in order to characterize the capacity of each primer to detect polymorphic loci among the genotypes. It is the sum total of the polymorphism information content (PIC) values of SSR markers produced by a particular primer. The PIC value was calculated using the formula PICi = 1 − ∑ P(i)2 [39], where pi is the frequency of the allele. The PIC values provided an estimate of the discriminatory power of any locus by considering the number of alleles per locus and the relative frequencies of these alleles in the population. The genetic diversity within the genotypes was calculated from the following equations and the PopGen program [40] using Nei’s gene diversity index [41] and the Shannon information index [42]. Structure 2.2 program was used to determine the genetic structure of genotypes [43]. In many genetic diversity studies with beans, genotypes are successfully divided into groups using the Structure program [44,45]. The F-statistics (FST) value reflects the difference between subpopulations [46]. Using the GenAlex program, basic coordinate analysis was carried out to better understand the diversity among genotypes. On the two-dimensional diagram obtained by covering the total variance of the first two coordinates, groups were determined and compared with cluster analysis. Genetic variation within and between populations was examined with the GenAlex program [47] using the analysis of molecular variance (AMOVA) method. Fst measures the amount of genetic variance that can be explained by population structure based on Wright’s F-statistics. An Fst value of 0 indicates no differentiation between the subpopulations while a value of 1 indicates complete differentiation [48]. In addition, genetic indices such as number of loci with private allele, number of different alleles (Na), number of effective alleles (Ne), Shannon’s information index (I), unbiased expected (uHe) and expected (He) for each proposed geographic region using the Genalex 6.5 software [45].

## 3. Results and Discussion

### 3.1. SSR Marker Information

Twenty-seven SSR markers produced a total of 142 bands, and the number of alleles per locus ranged from 2 (PVTT001) to 12 (BM153), with an average of 5.26. The polymorphism rate for each SSR marker was 100%. The allele frequency varied between 0.20 (BM153) and 0.78 (BM053). The lowest genetic diversity was 0.37 (BM053) and the highest 0.87 (BM153). Gene diversity is an important parameter used in the estimation of genetic variability between genotypes [41,49]. In a similar study, Dutta et al. [50] obtained a total of 150 alleles using 30 SSR markers in 52 Indian common bean genotypes. They found that the number of alleles ranged from 1 to 19 and the number of alleles per locus was 5. Investigating genetic variation in 60 Brazilian bean genotypes, [25] obtained 196 polymorphic alleles from 85 SSR markers and reported the average number of alleles per locus to range from 2 to 6, with an average of 2.8. Zhang et al. [51] investigated genetic diversity in 229 Chinese native bean genotypes using 30 SSR markers and the number of alleles varied between 2 and 19; they obtained an average of 5.5 alleles per locus and 116 alleles in total. The average PIC value obtained as a result of the analysis with the SSR marker, showing the discriminatory power of a marker, was 0.63, ranging between 0.33 (BM053) and 0.86 (BM153) depending on the markers. Having a high PIC value for a marker is one of the most important indicators that the marker can be used successfully in the evaluation of genetic variation [50]. Markers with high PIC values, such as BM141 (0.81), BMd1 (0.81), BM153 (0.86), and PVAT001 (0.81), are preferred in bean genetic diversity studies (Table 3). In other studies, with SSR markers in beans, the PIC value was 0.23–0.87 [51], 0.03–0.70 [25], 0–0.79 [37], 0.38–0.94 [50], 0.40–0.82 [52], and 0.42–0.88 [4]; these values varied widely and are consistent with our research results.

The number of effective alleles, which was 3.749 on average in the study, varied between 1.578 (BM053) and 7.864 (BM153) according to the markers (Table 4). The observed heterozygosity was 0.30 on average. The relatively low heterozygosity seen may be due to the autogamous structure of the bean [22,45,53]. However, compared with some other research results, e.g., Kyrgyzstan (0.05) [45], India (0.019) [54], and Brazil (0.16) [25], our value is higher. The expected heterozygosity, with a mean of 0.693, was lowest (0.370) at the BM053 locus and highest (0.882) at the BM153 locus (Table 4). Similar to our findings, [44] reported with 36 SSR markers in 104 wild bean genotypes that the mean expected heterozygosity value was 0.66, and the highest expected heterozygosity value (0.96) was obtained from the PVAT001 marker. Zargar et al. [46] determined the expected heterozygosity values as 0.2192 in the first subpopulation, 0.2124 in the second subpopulation, and 0.2821 in the third subpopulation, respectively, in their analysis using 15 RAPD and 23 SSR markers in 51 Indian bean genotypes. In this study, Shannon information index (I), which ranged from 0.663 (GATS91) to 2.202 (BM153), was found to be on average 1.343 (Table 4). The high Shannon knowledge index in our study showed that the SSR markers employed were useful in determining genetic diversity [55]. Gioia et al. [13] reported the Shannon information index to range from 0.19 to 0.74 (mean 0.66) with 58 SSR markers in 192 bean genotypes. In another study using 65 *Vigna umbellata* genotypes and 28 SSR markers, six geographical groups were formed, and the Shannon information index varied between 0.845 and 1.019 [56]. On the other hand, Öztürk et al. [2] investigated genetic diversity in 75 bean genotypes using 27 iPBS markers and found the Shannon information index to range from 0.570 to 0.636 (mean 0.599).

### 3.2. Cluster Analysis

Comparative analysis of molecular sequence data enables the determination of proximity or distance between genotypes and displays clusters of genotypes by constructing a phylogenetic tree. For this purpose, cluster analysis was performed among beans by the neighbor-joining method of the maximum composite likelihood substitution model, identifying three clustered groups. Considering the higher cophenetic correlation coefficient, the dendrogram was assumed to represent the similarity matrix very well. Cluster III consists of two subgroups; Aras-98, Elkoca-05, Göynük-98, Yakutiye-98 and Karacaşehir-90 cultivars were included in the first subcluster, along with five Ispir bean lines, and 63, 64, 65, 69 accessions were included in the second subcluster. In addition, cluster I consisted of two subgroups; there were twenty-three participants in the first subgroup and four participants in the second subgroup. In the cluster II, there were eight participants. (Figure 2 and Table 5). This clustering of genotypes showed that there was no significant relationship between geographic origin and genetic similarity. This result suggests that there may be some level of gene flow between genotypes or a recent introduction from a common source. In a similar study aiming to determine genetic diversity using 30 SSR markers in 50 bean genotypes, including 38 local bean genotypes obtained from the Northeast Anatolia region and 12 registered varieties, the genotypes clustered into two groups [57]. However, Öztürk et al. [2], who investigated genetic diversity by using 27 iPBS markers in 71 bean genotypes and 4 commercial varieties collected from Erzincan, determined that the genotypes clustered into two groups and both groups were further divided into two subgroups. In a study by [58], they conducted a genetic diversity study in beans using 26 iPBS primers. At the end of the research, it was determined that the bean inclusions were divided into three main clusters. However, while three subgroups were formed in our study, five subgroups were identified in the findings of the researchers.

Knowing the genetic distances between genotypes provides an enormous advantage in selecting suitable parents for bean breeding programs. In this study, the greatest genetic distance (0.83) was determined between the Karacaşehir-90 variety and Ispir bean lines 49, 50, and 53. The shortest genetic distance was observed between lines 6 and 26 and lines 27 and 28 (0.08 and 0.09, respectively) (Appendix A).

### 3.3. Determination of Genetic Diversity Based on Principal Coordinate Analysis

Principal coordinate analysis (PCoA) is a multivariate dataset that provides the ability to find and archive key patterns in multiple loci and multiple samples. With this technique, the distances between the groups, which are based on the two-dimensional diagram formed by the similarity or distance matrix between the individuals, reflect actual distances [59]. PCoA is used to provide a spatial representation of the relative genetic distances between populations [60].

In our study, the baseline coordinate analysis was performed using the neutral genetic distance of Nei. The percentage of genetic diversity explained by each of the three main coordinates of the basic coordinate analysis was 20.57, 16.96, and 13.33; together, these three components explained 50.85% of the diversity. Although the groups were not completely separated in the two-dimensional diagram obtained over the first two components, the distribution of genotypes on the diagram indicated the presence of genetic diversity (Figure 3). Genotypes taken from the Gaziler District of Ispir center are located on the upper left, the Maden village and Elmalı Town Ağıldere village genotypes on the lower left, and commercial varieties and Kirazlı village and Maden Köprübaşı Town Akbağ District genotypes on the lower right of Axis 1. The remaining genotypes are distributed over several sections of the diagram. For example, genotypes collected from Öztoprak village are clustered on the upper left and upper right sides of Axis 1, while Ulubel Village genotypes are distributed over all parts of the diagram. Yeşilyurt Village genotypes were located on the lower left sections of Axis 1 (Figure 3). This distribution of genotypes on the diagram shows that genetic diversity is weak both between commercial varieties and between Maden Village genotypes and Elmalı Town Ağıldere Village genotypes. Klaedtke et al. [60] reported that the baseline coordinate analysis they applied on 15 bean genotypes using the SSR marker grouped the genotypes in a meaningful way and the first two components explained 77.7% of the total variation. In their study using SSR markers in 349 wild and cultivated bean genotypes from the Andean and Mesoamerican gene pools, Kwak and Gepts [53] found that the results of basic coordinate analysis and genetic structure analysis were similar, and the first two components explained 66% of the total variation.

### 3.4. Molecular Variance Analysis

Analysis of molecular variance (AMOVA) revealed that within-population variance (66%) was higher than between-population variance (34%) (Table 6). This result indicates that there is gene flow between populations [45]. Blair et al. [44] in their study with 36 SSR markers in 104 wild bean genotypes determined that the variance within populations was 98%. Another study using 11 SSR markers in 28 bean genotypes grown in Kyrgyzstan [45] reported that the variance between populations was higher, contrary to our research findings. Rebaa et al. [55] who used 21 genotypes and 8 SSR markers in broad beans noted an intra-population variance of 83% and an inter-population variance of 17%, and their molecular variance analysis using SSR data revealed a significant intra-population genetic variation. Similarly, research results have been reported in which the within-population variance is higher than the between-population variance in different plant species such as apple [61] and lettuce [62].

The summary statistics for nine populations are listed in Table 7. We determined that the He value ranged from 0.038 (Ic) to 0.189 (Ov) (Mean 0.115), while the uHe value ranged from 0.051 (Ic) to 0.196 (Ov) (Mean 0.131). The I value among the nine populations ranged from 0.055 (Ic) to 0.290 (Ov) (Mean 0.173). The percentage of polymorphic loci (PPL) for bean was lowest at 9.15% (Ic) and 15.49% (Mka). Among the nine populations of bean, the PPL value ranged from 9.15% (Ic) to 64.08% (Ov) (Mean 33.41%). The Nei genetic (h) values of the nine bean populations are presented in Table 8. Among the nine populations of bean from Ispir, the smallest h values observed were in Ev/Mv (0.087), while the greatest were observed in Mka/Ic (0.341).

### 3.5. Genetic Structure Analysis

In many genetic diversity studies with beans, genotypes are successfully separated into groups using the structure program [44,45]. In this study, the population structure of accession in 45 bean genotypes was classified according to the SSR data, and three subpopulations were obtained with little mixing of genotypes regardless of geographical distribution (Figure 4). Geographical distribution is an important factor in terms of the genetic diversity of species [2,63]. In this study, the proximity of the places where the samples were collected can be counted as the reason for the mixing of these three populations [46]. The low number of populations in our study (K = 3) is due to the high rate of gene flow between the regions where the samples were taken [2,23]. According to these data, there are 17 local genotypes in the first subpopulation, 9 local genotypes together with the 5 commercial varieties in the second subpopulation, and 14 local genotypes in the third subpopulation (Table 9). Bean genotypes and geographic distributions of populations are presented in Figure 5. The F_ST_ (F-statistic) value was determined as 0.34, 0.26, and 0.41 in the first, second, and third subpopulations, respectively, and the mean F_ST_ (F-statistic) value of 0.34 confirmed the segregation of all subpopulations and the diversity into SSR alleles [44] (Table 10). Evaluating genetic diversity in 149 common bean genotypes using 24 SSR markers, Sharma et al. [54] determined that the genotypes were divided in three subpopulations. Zargar et al. [46] who performed genetic structure analysis and UPGMA clustering analysis on 51 Indian bean genotypes using 15 RAPD and 23 SSR markers stated that three groups were formed in both analyses. They emphasized that the FST values obtained as a result of the genetic structure analysis (0.4047, 0.3799, and 0.2059 for the 1st, 2nd, and 3rd subgroups, respectively) are a strong indicator of the effective separation of the subpopulations and the diversity in the SSR alleles. The results of the study reported by Khaidizar et al. [57] showed higher genetic polymorphism when they used SSR to investigate the level of polymorphism in Turkish common bean genotypes, which includes most of the genotypes used in Ceylan et al. [19] study. Consistent with several previous studies, cluster analysis revealed that it resulted in two major clusters, possibly representing two major gene pools, namely Andean and Mesoamerican. It was stated that these small-seeded cultivars, which clustered separately from the others in both plastid and nuclear marker analysis, may belong to the Mesoamerican gene pool.

## 4. Conclusions

Assessment of genetic variability of the germplasm is the first step, termed pre-breeding, for the improvement and development of superior cultivars. In the present study, genotypes collected from the Erzurum-Ispir region, located in the Northeastern Anatolia region of Türkiye, were evaluated at the molecular level. Our results showed a high level of genetic diversity within the population. It is important to collect local varieties and determine their genetic diversity in order to protect bean genetic resources and use them in breeding studies. An acquaintance of the genetic diversity and population structure of these genotypes may assist in the efficient management of these natural germplasms of beans. The results of this research have shown that the SSR marker system can be used successfully in determining genetic diversity among Ispir bean genotypes. These results are anticipated to guide the selection of appropriate markers in genetic diversity studies in beans.

## Figures and Tables

**Figure 1 genes-13-01410-f001:**
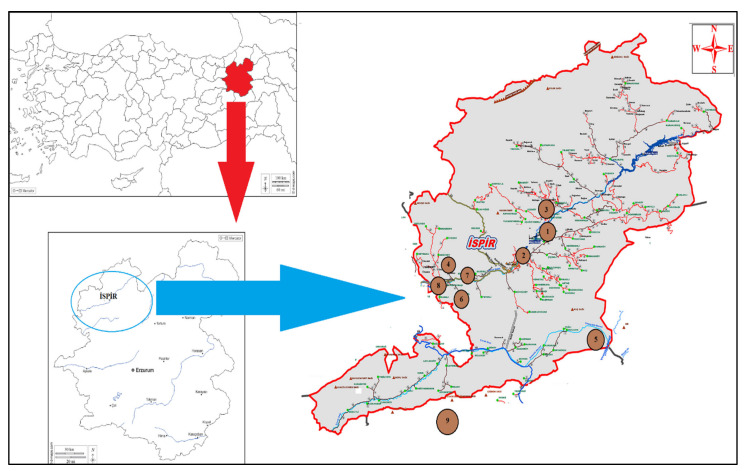
Geographic distribution of common bean landraces collected from different geographical provinces of Ispir, Türkiye (Table 1; 1: Öztoprak, 2: Ispir Center Gaziler Neighborhood, 3: Yeşilyurt Village, 4: Maden Village, 5: Elmalı District Ağıldere Village, 6: Ulubel Village, 7: Kirazlı Village, 8: Maden Köprübaşı District Akbağ Neighborhood, 9: Commercial varieties. Commercial varieties are shown off the map as they are not local genotypes).

**Figure 2 genes-13-01410-f002:**
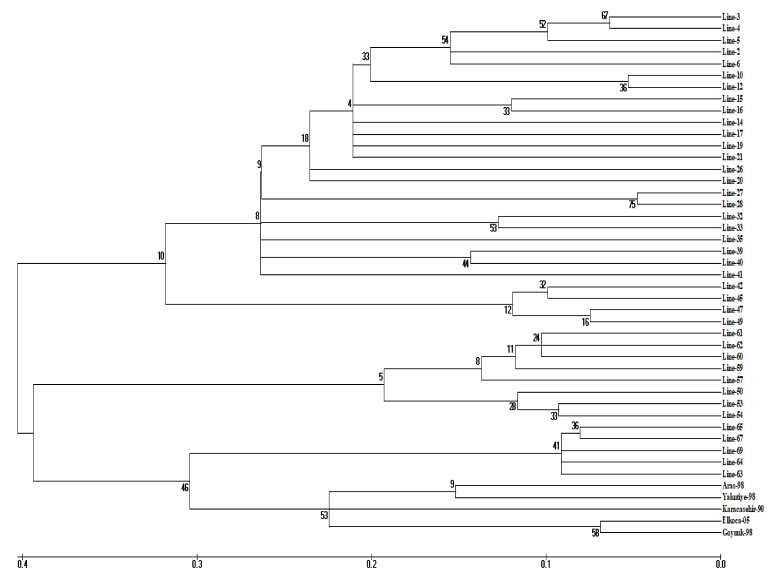
Dendrogram showing the genetic relationship between 45 bean genotypes generated by the neighbor-joining method of the MEGA software with the maximum composite likelihood substitution model using 27 SSR markers.

**Figure 3 genes-13-01410-f003:**
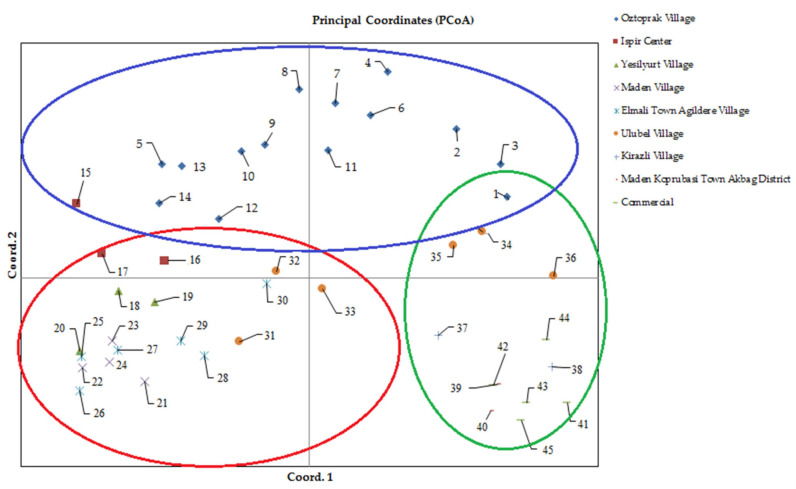
Principal coordinate analysis using SSR primer and separation on a two-dimensional diagram. The numbers in this figure represent the code numbers of the bean accessions presented in Table 1.

**Figure 4 genes-13-01410-f004:**
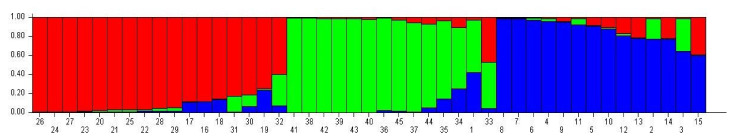
Graphic representation of population structure according to SSR data [For each bean genotype, subgroup 1 (red), subgroup 2 (green), and subgroup 3 (blue) are indicated by a vertical line representing the genotypes (K = 3)].

**Figure 5 genes-13-01410-f005:**
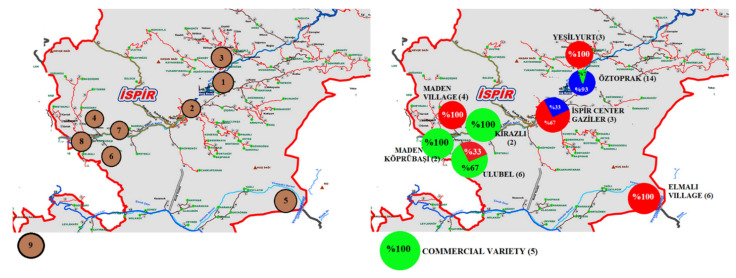
Geographical distribution of 45 bean genotypes identified by the STRUCTURE program using 27 SSR markers by region (1: Öztoprak, 2: Ispir Center Gaziler Neighborhood, 3: Yeşilyurt Village, 4: Maden Village, 5: Elmalı District Ağıldere Village, 6: Ulubel Village, 7: Kirazlı Village, 8: Maden Köprübaşı District Akbağ Neighborhood, 9: Commercial varieties. Commercial varieties are shown off the map as they are not local genotypes).

**Table 1 genes-13-01410-t001:** List of bean accession by information, coordinates, and some seed characteristics of the gathering place (Figure 1).

CN ^1^	ACN	Site-Location		Latitude	Longitude	Altitude (m)	SW (g)	SC	SS
1	Line-2	1-Öztoprak Village		40.518	41.052	1431	57.3	White	Circular
2	Line-3			40.518	41.052	1431	57.3	White	Oval
3	Line-4			40.518	41.052	1431	63.3	White	Circular
4	Line-5			40.518	41.052	1431	60.0	White	Oval
5	Line-6			40.518	41.052	1431	57.2	White	Circular
6	Line-10			40.518	41.052	1431	58.2	White	Circular
7	Line-12			40.518	41.052	1431	55.1	White	Oval
8	Line-14			40.518	41.052	1431	59.1	White	Circular
9	Line-15			40.518	41.052	1431	58.3	White	Circular
10	Line-16			40.518	41.052	1431	62.0	White	Circular
11	Line-17			40.518	41.052	1431	63.5	White	Circular
12	Line-19			40.518	41.052	1431	62.1	White	Circular
13	Line-20			40.518	41.052	1431	58.9	White	Oval
14	Line-21			40.518	41.052	1431	61.9	White	Circular
15	Line-26	2-Ispir Center Gaziler Neighborhood		40.485	41.002	1264	60.2	White	Circular
16	Line-27			40.468	40.983	1168	56.2	White	Circular
17	Line-28			40.468	40.983	1168	59.8	White	Circular
18	Line-32	3-Yeşilyurt Village		40.518	41.069	1549	57.0	White	Circular
19	Line-33			40.518	41.069	1549	57.7	White	Circular
20	Line-35			40.518	41.069	1549	58.1	White	Circular
21	Line-39	4-Maden Village		40.435	40.851	1226	54.5	White	Oval
22	Line-40			40.435	40.851	1226	54.1	White	Oval
23	Line-41			40.435	40.851	1226	55.9	White	Oval
24	Line-42			40.435	40.851	1226	56.6	White	Circular
25	Line-45	5-Elmalı District Ağıldere Village		40.401	40.834	1470	59.2	White	Circular
26	Line-47			40.401	40.834	1470	59.2	White	Circular
27	Line-49			40.401	40.834	1470	55.2	White	Circular
28	Line-50			40.401	40.834	1470	58.4	White	Circular
29	Line-53			40.401	40.834	1470	56.2	White	Circular
30	Line-54			40.401	40.834	1470	61.4	White	Circular
31	Line-57	6-Ulubel Village		40.418	40.868	1424	54.5	White	Circular
32	Line-59			40.418	40.868	1424	56.5	White	Circular
33	Line-60			40.418	40.868	1424	59.3	White	Circular
34	Line-61			40.418	40.868	1424	57.9	White	Circular
35	Line-62			40.418	40.868	1424	55.3	White	Circular
36	Line-63			40.418	40.868	1424	57.4	White	Circular
37	Line-64	7-Kirazlı Village		40.436	40.887	1220	58.2	White	Oval
38	Line-65			40.436	40.887	1220	54.8	White	Circular
39	Line-67	8-Maden Köprübaşı District Akbağ Neighborhood		40.434	40.819	1286	59.7	White	Circular
40	Line-69			40.434	40.819	1286	55.6	White	Circular
41	Aras-98	9-	Eastern Anatolia Agricultural Research Institute Directorate/Erzurum				45.4	White	Cylindrical
42	Elkoca-05		Ataturk University Faculty of Agriculture/Erzurum				49.6	White	Cylindrical
43	Göynük-98		Gateway Agricultural Research Institute Directorate/Eskişehir				53.5	White	Cylindrical
44	Karacaşehir-90		Gateway Agricultural Research Institute Directorate/Eskişehir				18.0	White	Oval
45	Yakutiye-98		Eastern Anatolia Agricultural Research Institute Directorate/Erzurum				43.9	White	Cylindrical

^1^ CN: Code number; ACN: Accession number; SW: 100-seed weight (g); SC: Seed color SS: Seed shape.

**Table 2 genes-13-01410-t002:** SSR primers and sequence information used for genetic diversity analysis among bean accessions.

Marker Name	GenBank Code	Size (bp)	Linkage Group	Motifs	Forward (5′–3′)	Reverse (5′–3′)	References
BMd1	X96999	165	Pv03	(AT)_9_	CAAATCGCAACACCTCACAA	GTCGGAGCCATCATCTGTTT	[33]
BMd15	K03288	166	Pv04	(ATGC)_4_	TTGCCATCGTTGCTTAATTG	TTGGAGGAAGCCATGTATGC	[33]
BMd18	X59469	216	Pv02	(GAAT)_3_	AAAGTTGGACGCACTGTGATT	TCGTGAGGTAGGAGTTTGGTG	[33]
BM053	AF324244	105	Pv01	(CT)21(CA)_19_(TA)_9_	TGCTGACCAAGGAAATTCAG	GGAGGAGGCTTAAGCACAAA	[33]
BM114	AF483854	234	Pv09	(TA)_8_(GT)_10_	AGCCTGGTGAAATGCTCATAG	CATGCTTGTTGCCTAACTCTCT	[34]
BM137	AF483855	155	Pv06	(CT)_33_	CGCTTACTCACTGTACGCACG	CCGTATCCGAGCACCGTAAC	[34]
BM141	AF483859	218	Pv09	(GA)_29_	TGAGGAGGAACAATGGTGGC	CTCACAAACCACAACGCACC	[34]
BM143	AF483861	143	Pv02	(GA)_35_	GGGAAATGAACAGAGGAAA	ATGTTGGGAACTTTTAGTGTG	[34]
BM152	AF483868	127	Pv02	(GA)_31_	AAGAGGAGGTCGAAACCTTAAATCG	CCGGGACTTGCCAGAAGAAC	[34]
BM153	AF483869	226	Pv08	(CA)_5_(TG)(CA)_3_CG(CA)_10_(TA)_4_	CCGTTAGGGAGTTGTTGAGG	TGACAAACCATGAATATGCTAAGA	[34]
BM154	AF483870	218	Pv09	(CT)_17_	TCTTGCGACCGAGCTTCTCC	CTGAATCTGAGGAACGATGACCAG	[34]
BM156	AF483872	267	Pv02	(CT)_32_	CTTGTTCCACCTCCCATCATAGC	TGCTTGCATCTCAGCCAGAATC	[34]
BM160	AF483876	211	Pv09	(GA)_15_(GAA)_5_	CGTGCTTGGCGAATAGCTTTG	CGCGGTTCTGATCGTGACTTC	[34]
BM161	AF483877	185	Pv04	(GA)_7_(GA)_8_	TGCAAAGGGTTGAAAGTTGAGAG	TTCCAATGCACCAGACATTCC	[34]
BM167	AF483881	165	Pv08	(GA)_19_	TCCTCAATACTACATCGTGTGACC	CCTGGTGTAACCCTCGTAACAG	[34]
BM175	AF483886	170	Pv05	(AT)_5_(GA)_19_	CAACAGTTAAAGGTCGTCAAATT	CCACTCTTAGCATCAACTGGA	[34]
PVAT001	U18791	239	Pv04	(AT)_22_	GGGAGGGTAGGGAAGCAGTG	GCGAACCACGTTCATGAATGA	[35]
PVAG004	X04660	201	Pv04	(AG)_8_	TTGATGACGTGGATGCATTGC	AAAGGGCTAGGGAGAGTAAGTTGG	[35]
PVBR14	DQ185881	196	Pv06	(AG)_23_	TGAGAAAGTTGATGGGATTG	ACGCTGTTGAAGGCTCTAC	[36]
BM183	AF483888	149	Pv07	(TC)_14_	CTCAAATCTATTCACTGGTCAGC	TCTTACAGCCTTGCAGACATC	[34]
BM188	AF483892	177	Pv09	(CA)_18_(TA)_7_	TCGCCTTGAAACTTCTTGTATC	CCCTTCCAGTTAAATCAGTCG	[34]
BM199	AF483896	304	Pv04	(GA)_15_	AAGGAGAATCAGAGAAGCCAAAAG	TGAGGAATGGATGTAGCTCAGG	[34]
BM200	AF483897	221	Pv01	(AG)_10_	TGGTGGTTGTTATGGGAGAAG	ATTTGTCTCTGTCTATTCCTTCCAC	[34]
BM210	AF483902	166	Pv07	(CT)_15_	ACCACTGCAATCCTCATCTTTG	CCCTCATCCTCCATTCTTATCG	[34]
BM211	AF483903	186	Pv08	(CT)_16_	ATACCCACATGCACAAGTTTGG	CCACCATGTGCTCATGAAGAT	[34]
GATS91	AF483842	229	Pv02	(GA)_17_	GAGTGCGGAAGCGAGTGAG	TCCGTGTTCCTCTGTCTGTG	[37]
PVTTTC01	X53603	163	Pv07	(GAAT)_5_	TGGACTCATAGAGGCGCAGAAAG	AAGGATGGGTTCCGTGCTTG	[35]

**Table 3 genes-13-01410-t003:** Summary information obtained with twenty-seven 27 SSR primer pairs used in bean accessions collected from İspir location.

Locus	Allele Number	Number of Polymorphic Bands	Polymorphism Percentage (%)	Allele Frequency	Gene Diversity	Observed Heterozygosity	PIC *
BM053	3	3	100	0.78	0.37	0.00	0.33
BM114	6	6	100	0.36	0.74	0.93	0.69
BM137	6	6	100	0.29	0.78	0.00	0.75
BM141	7	7	100	0.22	0.83	0.98	0.81
BM143	11	11	100	0.29	0.78	0.96	0.75
BM152	6	6	100	0.33	0.75	1.00	0.71
BM153	12	12	100	0.20	0.87	0.51	0.86
BM154	4	4	100	0.67	0.48	0.00	0.42
BM156	5	5	100	0.30	0.77	1.00	0.73
BM160	7	7	100	0.32	0.77	0.96	0.73
BM161	3	3	100	0.49	0.57	0.00	0.48
BM167	6	6	100	0.28	0.80	0.00	0.77
BM175	3	3	100	0.38	0.66	0.00	0.59
BM183	3	3	100	0.59	0.50	0.00	0.40
BM188	5	5	100	0.44	0.67	0.00	0.61
BM199	3	3	100	0.42	0.64	0.00	0.56
BM200	4	4	100	0.42	0.68	0.87	0.62
BM210	6	6	100	0.47	0.71	0.00	0.67
BM211	6	6	100	0.31	0.79	0.00	0.76
BMd1	7	7	100	0.22	0.83	1.00	0.81
BMd15	4	4	100	0.36	0.73	0.00	0.68
BMd18	3	3	100	0.58	0.56	0.00	0.49
GATS91	2	2	100	0.62	0.47	0.00	0.36
PVAG004	6	6	100	0.49	0.69	0.00	0.65
PVAT001	8	8	100	0.24	0.83	0.00	0.81
PVBR14	4	4	100	0.31	0.74	0.00	0.69
PVTT001	2	2	100	0.60	0.48	0.00	0.36
Mean	5.26	5.26	100	0.41	0.69	0.30	0.63

* PIC: polymorphism information content.

**Table 4 genes-13-01410-t004:** Effective number of alleles (Ne), expected heterozygosity (He) and Shannon information index (I) based on 27 SSR loci bean accessions.

Marker	Ne *	He **	I ***	Marker	Ne *	He **	I ***
BM053	1.578	0.370	0.665	BM188	3.017	0.676	1.223
BM114	3.824	0.746	1.472	BM199	2.770	0.646	1.054
BM137	4.633	0.793	1.614	BM200	3.129	0.688	1.243
BM141	5.921	0.84	1.832	BM210	3.403	0.714	1.450
BM143	4.581	0.790	1.752	BM211	4.856	0.803	1.668
BM152	4.074	0.763	1.515	BMd1	6.026	0.843	1.841
BM153	7.864	0.882	2.202	BMd15	3.708	0.738	1.349
BM154	1.930	0.487	0.846	BMd18	2.298	0.571	0.944
BM156	4.336	0.778	1.529	GATS91	1.887	0.475	0.663
BM160	4.281	0.775	1.619	PVAG004	3.199	0.695	1.420
BM161	2.351	0.581	0.9291	PVAT001	5.973	0.841	1.895
BM167	4.930	0.806	1.6542	PVBR14	3.785	0.744	1.356
BM175	2.939	0.667	1.088	PVTT001	1.916	0.483	0.671
BM183	2.004	0.506	0.764	Mean	3.749	0.693	1.343

* Ne: effective number of alleles, ** He: expected heterozygosity and *** I: Shannon’s information index, respectively.

**Table 5 genes-13-01410-t005:** Groups and subgroups of *P. vulgaris* accessions determined as a result of Neighbor Joining (NJ) cluster analysis.

Group	Subgroup	Genotype	Total Genotype Number
I	1	2, 3, 4, 5, 6, 10, 12, 14, 15, 16, 17, 19, 20, 21, 26, 27, 28, 32, 33, 35, 39, 40, 41	23
	2	42, 45, 47, 49	4
II	-	50, 53, 54, 57, 59, 60, 61, 62	8
III	1	63, 64, 65, 67, 69	5
	2	Aras-98, Elkoca-05, Göynük-98, Yakutiye-98, Karacaşehir-90	5

**Table 6 genes-13-01410-t006:** Analysis of molecular variance (AMOVA) among and within the bean genotypes, using SSR marker.

Source	Degree of Freedom (DF)	Sum of Squares (SS)	Variance Component	% of Total Variance	*p-*Value
Among Population	8	410.408	7.876	34%	0.339
Within Population	36	537.774	15.365	66%	0.001
Total	44	948.182	23.241	100%	

**Table 7 genes-13-01410-t007:** Estimates of genetic diversity and distribution of gene diversity between the populations of *P. vulgaris* L. assessed with twenty-seven SSR primers.

Population	N	Na	Ne	I	He	uHe	PPL (%)
Ov	14	1.289	1.311	0.290	0.189	0.196	64.08%
Ic	3	0.387	1.065	0.055	0.038	0.051	9.15%
Yv	3	0.697	1.197	0.168	0.114	0.137	29.58%
Mv	4	0.831	1.212	0.193	0.127	0.145	37.32%
Eav	6	0.880	1.242	0.215	0.143	0.156	40.85%
Uv	6	0.845	1.228	0.202	0.134	0.147	38.73%
Kv	2	0.500	1.120	0.102	0.070	0.093	16.90%
Mka	2	0.472	1.110	0.094	0.064	0.086	15.49%
Com	5	1.014	1.259	0.238	0.156	0.173	48.59%
Mean		0.768	1.194	0.173	0.115	0.131	33.41%

N: number of sample size, Na: number of distinct alleles, Ne: effective number of alleles, I: Shannon’s information index, He: expected heterozygosity, uHe: unbiased expected heterozygosity, PPL: percentage of polymorphic loci; Ov: Oztoprak Village; Ic: Ispir Center, Yv: Yesilyurt Village, Mv: Maden Village, Eav: Elmali Town Agildere Village, Uv: Ulubel Village, Kv: Kirazli Village; Mka: Maden Koprubasi Town Akbag District, Com: Commercial.

**Table 8 genes-13-01410-t008:** Pairwise population matrix of Nei genetic (h) distance for nine groups of beans (*P. vulgaris* L.) genotypes.

	Ov	Ic	Yv	Mv	Ev	Uv	Kv	Mka	Com
Ov	0.000								
Ic	0.170	0.000							
Yv	0.124	0.134	0.000						
Mv	0.129	0.153	0.089	0.000					
Ev	0.118	0.196	0.155	0.087	0.000				
Uv	0.098	0.245	0.201	0.177	0.111	0.000			
Kv	0.195	0.309	0.278	0.234	0.213	0.113	0.000		
Mka	0.195	0.341	0.258	0.225	0.237	0.172	0.111	0.000	
Com	0.125	0.283	0.190	0.166	0.181	0.154	0.178	0.128	0.000

Ov: Oztoprak Village; Ic: Ispir Center, Yv: Yesilyurt Village, Mv: Maden Village, Eav: Elmali Town Agildere Village, Uv: Ulubel Village, Kv: Kirazli Village; Mka: Maden Koprubasi Town Akbag District, Com: Commercial.

**Table 9 genes-13-01410-t009:** Membership coefficient of bean genotypes in three subpopulations.

No.	Genotype	Subpopulation			No.	Genotype	Subpopulation		
		I	II	III			I	II	III
1	Line-2	0.026	0.551 *	0.422	24	Line-42	0.987 *	0.007	0.006
2	Line-3	0.009	0.216	0.775 *	25	Line-45	0.970 *	0.016	0.015
3	Line-4	0.008	0.349	0.643 *	26	Line-47	0.991 *	0.005	0.004
4	Line-5	0.01	0.027	0.963 *	27	Line-49	0.987 *	0.006	0.007
5	Line-6	0.085	0.005	0.910 *	28	Line-50	0.951 *	0.029	0.02
6	Line-10	0.007	0.017	0.975 *	29	Line-53	0.947 *	0.034	0.019
7	Line-12	0.006	0.006	0.988 *	30	Line-54	0.808 *	0.124	0.068
8	Line-14	0.005	0.004	0.99 *	31	Line-57	0.826 *	0.161	0.013
9	Line-15	0.04	0.01	0.95 *	32	Line-59	0.599 *	0.323	0.078
10	Line-16	0.102	0.018	0.88 *	33	Line-60	0.466	0.488 *	0.046
11	Line-17	0.014	0.064	0.922 *	34	Line-61	0.102	0.644 *	0.254
12	Line-19	0.167	0.025	0.808 *	35	Line-62	0.035	0.820 *	0.145
13	Line-20	0.208	0.007	0.785 *	36	Line-63	0.006	0.970 *	0.024
14	Line-21	0.218	0.009	0.773 *	37	Line-64	0.057	0.931 *	0.013
15	Line-26	0.387	0.006	0.607 *	38	Line-65	0.006	0.989 *	0.005
16	Line-27	0.880 *	0.005	0.115	39	Line-67	0.01	0.983 *	0.007
17	Line-28	0.885 *	0.006	0.108	40	Line-69	0.02	0.974 *	0.006
18	Line-32	0.853 *	0.01	0.137	41	Aras-98	0.005	0.990 *	0.004
19	Line-33	0.746 *	0.016	0.238	42	Elkoca-05	0.01	0.984 *	0.006
20	Line-35	0.977 *	0.01	0.013	43	Göynük-98	0.012	0.982 *	0.007
21	Line-39	0.971 *	0.018	0.012	44	Karacaşehir-90	0.069	0.879 *	0.053
22	Line-40	0.969 *	0.015	0.016	45	Yakutiye-98	0.028	0.957 *	0.015
23	Line-41	0.980 *	0.007	0.014					

* in the table used to highlight which subgroup the participants belong to.

**Table 10 genes-13-01410-t010:** Expected heterozygosity and FST values in bean subpopulations.

Subpopulation (K)	Expected Heterozygosity	F_ST_
1	0.243	0.34
2	0.269	0.26
3	0.228	0.41
Mean	0.247	0.34

## Data Availability

Data is contained within the article.

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
