# Peer review of "Determining Genetic Diversity and Population Structure of Common Bean (Phaseolus vulgaris L.) Landraces from Türkiye Using SSR Markers"

_genes, 2022, doi:10.3390/genes13081410_

Round 1

Reviewer 1 Report

Dear Authors,

Reviewer comments genes-1809872

The manuscript entitled „Determining genetic diversity and population structure of common bean (Phaseolus vulgaris L.) landraces from Türkiye using SSR markers“ represents a valuable study on genetic diversity in the set of 45 common bean landraces from Anatolia region in Türkiye and 5 nationally registered genotypes using SSR genetic markers. The authors identified a set of 27 SSR markers and provided a detailed genetic analysis of the SSR markers used as well as multivariate statistical analyses such as cluster analysis and principal coordinate analysis of the studied genotypes of common bean based on the SSR markers characteristics.

I think that the present manuscript represents an important and valuable contribution to study genetic diversity in common bean gene pool thus I can recommend the publication of the present study in Genes.

I have only two major comments and some minor formal comments on the present manuscript:

Major comments:

1/ Terminology: The authors use several terms in Results section related to the analysis of SSR markers which are, however, not explained in Materials and methods, section 2.4. Molecular data analysis. All terms used in Results and related to SSR markers analysis which may not be familiar to some readers should be explained in Materials and methods, section 2.4. Molecular data analysis. The terms „He, expected heterozygosity“, „uHe, unbiased expected heterozygosity“, „ENS, effective number of alleles“, „FST values“ (Table 11) „I, Shannon information index“ (defined by providing an appropriate reference), „Nei´s gene diversity index“ (defined by providing an appropriate reference). In addition, the authors use two difefrent symbols for expecetd heterozygosity, „Eh“ and „He“ – the terminology should either be unified or the differences if they really exist have to be explained in the text in Materials and methods. I think that in addition to providing the references, brief explanation and formulas for calculation of the terms used in Results has to be provided in the section 2.4.

2/ Results: Cluster analysis – information provided in Figure 2 and Table 5: I think that the numbering of groups (clusters) I and III provided in Figure 2 and Table 5 should be reversed, i.e., the description of group (cluster) I provided in Table 5 corresponds to Cluster III in Figure 2, and vice versa, i.e., group III in Table 5 corresponds to cluster I in Figure 2. The authors should check it.

Formal comments on the text:

Terminology:

In Introduction, line 52 and further in the text, the authors use the term „gene centers“ for regions which represent the centers of genetic diversity for common bean. I think that instead of „gene center“, the term „center of genetic diversity“ has to be used in the text.

Further formal comments:

Abstract, line 41: Add „a“ preceding the words „genetic resource“, i.e., „The result confirmed that the rich diversity existing in Ispir bean landraces could be used as a genetic resource…“

Line 339: Replace the verb „were collected“ with „were divided“ in the statement „Sharma et al.   Determiend that the genotypes were divided in three subpopulations.“

Final recommendation: Reconsider after a major revision.

Author Response

 Responses to Comments of Reviewer 1

General Response:

Dear reviewer; We tried to respond to your valuable suggestions and comments in the best way possible. I hope it was a successful arrangement. The edits you want are highlighted in yellow background color.

Sincerely

Dr. Aras Turkoglu, Dr. Peter Poczai et al.

Comment

1.       Terminology: The authors use several terms in Results section related to the analysis of SSR markers which are, however, not explained in Materials and methods, section 2.4. Molecular data analysis. All terms used in Results and related to SSR markers analysis which may not be familiar to some readers should be explained in Materials and methods, section 2.4. Molecular data analysis. The terms „He, expected heterozygosity“, „uHe, unbiased expected heterozygosity“, „ENS, effective number of alleles“, „FST values“ (Table 11) „I, Shannon information index“ (defined by providing an appropriate reference), „Nei´s gene diversity index“ (defined by providing an appropriate reference). In addition, the authors use two difefrent symbols for expecetd heterozygosity, „Eh“ and „He“ – the terminology should either be unified or the differences if they really exist have to be explained in the text in Materials and methods. I think that in addition to providing the references, brief explanation and formulas for calculation of the terms used in Results has to be provided in the section 2.4.

Response:

Added text with necessary explanations to the relevant section…… “Fst measures the amount of genetic variance that can be explained by population structure based on Wright’s F-statistics, while Nm = [(1/Fst) − 1]/4. An Fst value of 0 indicates no differentiation between the subpopulations while a value of 1 indicates complete differentiation (Bird et al., 2017). In addition, genetic indices such as number of loci with private allele, number of different alleles (Na), number of effective alleles (Ne), Shannon’s information index (I), unbiased expected (uHe) and expected (He) for each proposed geographic region using the Genalex 6.5 software [45].”

In addition, the ambiguity between parameters with the same meaning specified with separate abbreviations has been removed. For example; Instead of “ENS”, the abbreviation “Ne” is used throughout the text.

Comment

2.       Results: Cluster analysis – information provided in Figure 2 and Table 5: I think that the numbering of groups (clusters) I and III provided in Figure 2 and Table 5 should be reversed, i.e., the description of group (cluster) I provided in Table 5 corresponds to Cluster III in Figure 2, and vice versa, i.e., group III in Table 5 corresponds to cluster I in Figure 2. The authors should check it.

Response: Necessary checks have been made. Group 1 and Group 3 have been replaced in Table 5. It became compatible with Figure 2.

Comment

3.       Terminology:

In Introduction, line 52 and further in the text, the authors use the term „gene centers“ for regions which represent the centers of genetic diversity for common bean. I think that instead of „gene center“, the term „center of genetic diversity“ has to be used in the text.

Response: The sentence has been corrected according to your suggestion.

“These centers of genetic diversity are separated from each other……………”

Comment

4.       Abstract, line 41: Add „a“ preceding the words „genetic resource“, i.e., „The result confirmed that the rich diversity existing in Ispir bean landraces could be used as a genetic resource…“

Response: The sentence has been corrected according to your suggestion.

“….landraces could be used as a genetic resource in designing……”

Comment

5.       Line 339: Replace the verb „were collected“ with „were divided“ in the statement „Sharma et al.   Determiend that the genotypes were divided in three subpopulations.“

Response: : The sentence has been corrected according to your suggestion.

“……determined that the genotypes were divided in three subpopulations…….”

Reviewer 2 Report

The paper title for the manuscript “Determining Genetic Diversity and Population Structure of 2 Common Bean (Phaseolus vulgaris L.) Landraces from Türkiye Using SSR Markers” provides an apt description of the research. The study consisted of 40 accessions representing landrace materials collected from a localized region in NE Turkey, plus 5 registered commercial varieties. The commercial lines are registered nationally and it is not apparent whether they also originated from the same region, or elsewhere in the country. After genotyping with 27 SSR markers, a number of genetic diversity metrics are reported. The main story is that these accessions could be grouped into three clusters based on genotypic similarities. All of the commercial lines grouped within one of the clusters. Yes, there is genetic variation among accessions, but the genotypic information by itself does not provide a complete picture. As such, this work represents one small work of rather limited interest and value to anyone outside of breeding programs in Turkey. This work really needs to be placed within the larger picture of bean introductions from the New World into the Old World. Some questions to consider would be what centers/races were introduced? Has there been much admixture among different materials? Were there many or only a few introductions?

 The authors are aware that different centers of domestication (“gene centers” in their manuscript) and races of bean exist based on their citations of the literature. However, there are no genotypes of known race/center of domestication that provide context as to which of the clusters belong to which race/center. In this case, phenotypic data could be used to assign races/centers. To start, passport data on the accessions should be included as a supplemental table. Characteristics that would help determine race would include seed size and shape, bracteole shape and size, leaf shape and size as well as several others  and it would be most helpful if phenotypic data were included in this study. I suspect that the 2 centers (Middle American and Andean) and 3 races (Mesoamerica, Durango and Nueva Granada) are present and seed size alone might distinguish among the three. Based on historical observations, race Durango, represented by medium-sized white beans, is common in Turkey, but its not possible to determine which of the clusters in the present work this might be. Seed color would be another characteristic of interest, and it would be interesting to document whether other colors are found among landraces and whether white color predominates. Another useful characteristic would phaseolin seed storage protein allele. Has this set of material been characterized using SDS-PAGE of seed proteins? If not, it may be possible that one or more of the SSR markers is linked to the phaseolin locus and allele could be inferred based on SSR allele present.

 In the introduction, the authors should include a short paragraph about the introduction of beans into Turkey after the beginning of the Colombian exchange. This could provide some context as to when and what kinds of beans were introduced into Turkey.

 The authors have failed to cite and discuss some relevant literature. In particular, a recently published paper uses the same set of accessions, but different markers to examine genetic diversity (HaliloÄŸlu, K., TürkoÄŸlu, A., Öztürk, H. I., Özkan, G., Elkoca, E., & Poczai, P. (2022). iPBS-Retrotransposon Markers in the Analysis of Genetic Diversity among Common Bean (Phaseolus vulgaris L.) Germplasm from Türkiye. Genes, 13(7), 1147.) The authors need to compare the findings in this study with those in the present work. This recently published paper also suffers from a lack of context in relation to the larger picture.

 There are two potential issues with the analysis. First, are the SSR markers used in this study unlinked? This is one of the assumptions made in genetic diversity studies and it should be possible to determine whether this assumption has been violated because most if not all of the markers used in this study have been mapped. It should not be difficult to determine to which linkage group they have been assigned using the list of bean SSR primers found on the BIC genetic webpage: http://www.bic.uprm.edu/?page_id=91. Original and subsequent publications using these markers will also have linkage information and can be consulted if markers do locate to the same linkage group. The authors should include the linkage group in their table of SSR markers and should address the issue of linkage of markers in the results and discussion. They may need to thin out any closely linked markers and redo the genetic diversity analysis.

Secondly, UPGMA was used to construct a dendrogram for the bean accessions. UPGMA suffers from being ultrametric and can lead to erroneous trees. Neighbor Joining is a method that does not have this problem and should be used instead of UPGMA. Also, there is no attempt to determine what is the “best” tree. Usually, bootstrapping is performed to determine how frequently the various nodes of the dendrogram are represented, and these are indicated as numbers (out of x trees) or percent at the nodes of the dendrogram. Other means of determining confidence in dendrograms exist, and the authors need to provide a statistical basis for the choice of the most representative dendrogram.

 In the discussion, the authors should discuss admixture in Turkish beans in relation to what has been found elsewhere in Europe. The amount of mixing across the European continent among races and centers of domestication is very high compared to the Americas (Angioi, S. A., Rau, D., Attene, G., Nanni, L., Bellucci, E., Logozzo, G., et al. (2010). Beans in Europe: origin and structure of the European landraces of Phaseolus vulgaris L. Theor. Appl. Genet. 121, 829–843. doi: 10.1007/s00122-010-1353-2). The question is whether a similar level of admixture has been observed in Turkey, or has the racial structure been preserved?

 Some specific comments in the manuscript follow.

 L24: Why are SSRs the “best” markers for studying genetic diversity? SNPs are codominant, biallelic and much more abundant than SSRs.

L49: Beans are not “the” primary source vegetable protein, but one of many. Change “the” to “a”.

 L51: Is [3] the appropriate reference for this statement? It references a paper on genetic diversity (not nutrition) in Serbian beans.

 L53: Use “Middle American” rather than “Central American” or “Mesoamerican” for center of domestication. Also, replace “gene centers” with the term “centers of domestication” which is much more descriptive. And to be clear, “Middle American” is used to describe the center of domestication to prevent confusion when discussing race “Mesoamerica”.

 L54: Check that reference numbers are correct. [4] is about genetic diversity in Croatian beans but I think the information in this sentence is better supported by [5]. See also [7] which describes genetic diversity techniques and doesn’t mention beans at all.

 L59: Seeds of genotypes should be seeds of phenotypes

 L66-68: This statement about narrowing of the genetic base cites a paper from Brazil. Is there a more general reference?

 L71: Rather than saying “the gene center for many plant species”, I would say “the center of domestication for important Old World cereal and grain legume crops”

 L95-96: I don't agree with this statement. The Illumina Beadchip is widely available & cost is ~$20/sample for the 12K chip, which is the cheapest of high-throughput technologies available. Its disadvantage is that you have to run all the SNPs, which is overkill if you are only interested in a specific chromosome region. GBS is more expensive but still under $50/sample for ~50K SNPs.

 L123: Tables and figures in this paper generally do not have complete titles. Figures and tables should be able to be interpreted without reading the text, so table titles and figure legends should be detailed, and footnotes used to explain abbreviations etc. In the case of Table 1, its not clear what country/region these bean accessions come from and that they are being used for a genetic diversity study. Also, in table 1, its difficult to know which sets of accessions to which site numbers correspond. The site number should be in line with the first accession of its group.

 L129: What are the modifications made to DNA extraction?

 L149: Need complete title for table 2. Also, there are some discrepancies in the table: Markers BM114 to BM175 belong to reference 34, not 32; Marker PV-AG004 belongs to reference 31, not 32. Did the forward and reverse sequences change in the PVBR14 marker? If so, what is the reason? Also, according to the cited source, the same marker was not used for the reverse primer (according to the reference DQ185880 was used) even though the forward primer belonged to the PVBR14 marker (there is a possibility that it will be PVBR20). If changes are to be made from the sources used, these changes should be stated. Has the reverse sequence of the BM153 marker been modified? The forward sequence of the GATS91 marker belongs to reference 34, not 32. The reverse sequence is not available in the source. PVTTTC001 should be PV-TTTC001 and belongs to reference 31, not 35. I could not find the first 4 markers in the given references.

 L180: Allele frequency ranged from 0.20 for BM153 to 0.78 for BM053 according to table 3 – rectify with the text.

 L200: Table 3 title needs to describe the crop, region and purpose of the study. Also, in Table 3, I am surprised to see such high levels of heterozygosity for some SSR markers since beans are highly self-pollinated.

 L231: Cluster A is cluster III in Fig 2 and group I in Table 5. Naming these three the same will clear up the confusion. In cluster A, it is said that there are 4 Ispir lines in the first subcluster. However, 5 types of Ispir lines are seen in Fig 2 and Table 5.

 L236-237: What appears to be gene flow among genotypes might be recent introduction from a common source.

 L248-249: The statement about knowing genetic distances helps select suitable parents in a breeding program needs to be qualified. If you chose parents based on maximum genetic diversity, you will be crossing between centers of domestication and you are not likely to produce elite lines from such crosses because coadapted center-specific complexes of genes are recombined and results in mostly lines with poor performance. If you want elite lines for release, it is much better to cross within centers of domestication, but the diversity data would allow you to choose parents that are least similar.

 L250-251: How did you determine genetic distance? There is no reference to a table or figure here and it is not possible to derive this information from Fig 2.

L266: There are three samples (code #s 15, 16, and 17) taken from the Gaziler district. However, there are only two samples(15 and 16) in Fig 3. Was 17 not analyzed? Also, although sample 17 belongs to the Gaziler district, it appears to belong to the Yesilyurt village in Fig 3. I think there is slippage in the examples. If not, this difference should be explained.

 Line 272: Sample 31, which belongs to the Ulubel village, is seen in the upper left side. 

 Line 273: YeÅŸilyurt village examples can be seen only in the lower left side. Fig 3, not 2.

 Line 274-275: Comparisons were made between samples representing only 3 regions. What is the reason of this?

 L283: Data in Table 6 is adequately described in the text and can be deleted.

L 284: Fig 3: Do the numbers given in the figure represent the code numbers of the samples given in Table 1? Necessary explanation should be given under the figure. Although a total of 45 accessions were used in the study, there are just 44 accessions in Fig 3.

 L303: Text differs from table - The I value ranged from 0.055 (Ic) to 0.290 (mean 0.173) in Table 8.

 Line 306: PPL mean value is 33.41% in Table 8.

 Line 333: The expression states that there are 14 genotypes in the third subgroup. Specify if example 5 (Line-6) belongs to the third subgroup.

 The figure on page 13 and line 327 should be Fig 4.

 The figure on page 14 and line 335 should be Fig 5.

 Figure 5 (former Fig 4) seems to be based on PCoA rather than structure analysis. Also, red and green colors in the pie charts in this figure are difficult for those with red-green color blindness to see. The red and green colors in the STRUCTURE analysis figure are ok.

 L350: Table 10 – what do the bolded number mean?

Author Response

Responses to Comments of Reviewer 2

General Response:

Dear reviewer; We tried to respond to your valuable suggestions and comments in the best way possible. I hope it was a successful arrangement. The edits you want are highlighted in blue background color.

Sincerely

Dr. Aras Turkoglu, Dr. Peter Poczai et al.

1.       The authors are aware that different centers of domestication (“gene centers” in their manuscript) and races of bean exist based on their citations of the literature. However, there are no genotypes of known race/center of domestication that provide context as to which of the clusters belong to which race/center. In this case, phenotypic data could be used to assign races/centers. To start, passport data on the accessions should be included as a supplemental table. Characteristics that would help determine race would include seed size and shape, bracteole shape and size, leaf shape and size as well as several others  and it would be most helpful if phenotypic data were included in this study. I suspect that the 2 centers (Middle American and Andean) and 3 races (Mesoamerica, Durango and Nueva Granada) are present and seed size alone might distinguish among the three. Based on historical observations, race Durango, represented by medium-sized white beans, is common in Turkey, but its not possible to determine which of the clusters in the present work this might be. Seed color would be another characteristic of interest, and it would be interesting to document whether other colors are found among landraces and whether white color predominates. Another useful characteristic would phaseolin seed storage protein allele. Has this set of material been characterized using SDS-PAGE of seed proteins? If not, it may be possible that one or more of the SSR markers is linked to the phaseolin locus and allele could be inferred based on SSR allele present.

In the introduction, the authors should include a short paragraph about the introduction of beans into Turkey after the beginning of the Colombian exchange. This could provide some context as to when and what kinds of beans were introduced into Turkey.

Response:

There was a limited number of seeds of genotypes collected from locations in this study. Therefore, parameters such as phenotypic characteristics and seed protein analysis were not investigated. We are planning to carry out characterization studies based on morphological, phenotypic and biochemical characteristics after the seeds of the genotypes are multiplied. Since we do not have these parameters, the following sentence has been added to the introduction part, taking into account your suggestion:

“It is thought that the bean was introduced to Europe during the 1st voyage of Columbus from Europe in the 16th and 17th centuries, and it was introduced to Türkiye in the 17th century, allowing the bean genotypes to spread to different parts of the world [Aydin and Baloch 2019].”

3.       The authors have failed to cite and discuss some relevant literature. In particular, a recently published paper uses the same set of accessions, but different markers to examine genetic diversity (HaliloÄŸlu, K., TürkoÄŸlu, A., Öztürk, H. I., Özkan, G., Elkoca, E., & Poczai, P. (2022). iPBS-Retrotransposon Markers in the Analysis of Genetic Diversity among Common Bean (Phaseolus vulgaris L.) Germplasm from Türkiye. Genes, 13(7), 1147.) The authors need to compare the findings in this study with those in the present work. This recently published paper also suffers from a lack of context in relation to the larger picture.

Response:

In the discussion part, a comparison was made with the related article. The following sentence has been added:

“In a study by Haliloglu et al (2022), they conducted a genetic diversity study in beans using 26 iPBS primers. At the end of the research, it was determined that the bean inclusions were divided into 3 main clusters. Similar results were obtained in our study as well. However, while 3 subgroups were formed in our study, 5 subgroups were identified in the findings of the researchers. This difference can be explained by the different markers used in the studies”

4.       -There are two potential issues with the analysis. First, are the SSR markers used in this study unlinked? This is one of the assumptions made in genetic diversity studies and it should be possible to determine whether this assumption has been violated because most if not all of the markers used in this study have been mapped. It should not be difficult to determine to which linkage group they have been assigned using the list of bean SSR primers found on the BIC genetic webpage: http://www.bic.uprm.edu/?page_id=91. Original and subsequent publications using these markers will also have linkage information and can be consulted if markers do locate to the same linkage group. The authors should include the linkage group in their table of SSR markers and should address the issue of linkage of markers in the results and discussion. They may need to thin out any closely linked markers and redo the genetic diversity analysis.

-Secondly, UPGMA was used to construct a dendrogram for the bean accessions. UPGMA suffers from being ultrametric and can lead to erroneous trees. Neighbor Joining is a method that does not have this problem and should be used instead of UPGMA. Also, there is no attempt to determine what is the “best” tree. Usually, bootstrapping is performed to determine how frequently the various nodes of the dendrogram are represented, and these are indicated as numbers (out of x trees) or percent at the nodes of the dendrogram. Other means of determining confidence in dendrograms exist, and the authors need to provide a statistical basis for the choice of the most representative dendrogram.

Response:

-Since phenotypic features were not used in this study, mapping was not performed. We will consider your suggestion in similar studies on phenotypic features.

- In this study, three different genetic distances (Jaccard, Dice and Simple Matching) and three different clustering methods (UPGMA, Single Linkage and Complete Linkage) were tested. As a result of the test, the highest cophenetic coefficient was selected for genetic distance and clustering method (Dice and UPGMA, r=0.745).  In addition, the following sentence was added to the tresult section:

“In this study, the cophenetic correlation coefficient between the similarity matrix and the UPGMA dendrogram in SSR analyzes was found to be significantly high (r=0.745). Con-sidering the higher cophenetic correlation coefficient, the dendrogram was assumed to represent the similarity matrix very well.”

5.       In the discussion, the authors should discuss admixture in Turkish beans in relation to what has been found elsewhere in Europe. The amount of mixing across the European continent among races and centers of domestication is very high compared to the Americas (Angioi, S. A., Rau, D., Attene, G., Nanni, L., Bellucci, E., Logozzo, G., et al. (2010). Beans in Europe: origin and structure of the European landraces of Phaseolus vulgaris L. Theor. Appl. Genet. 121, 829–843. doi: 10.1007/s00122-010-1353-2). The question is whether a similar level of admixture has been observed in Turkey, or has the racial structure been preserved?

Response:  There are studies on whether a similar level of contribution is observed in Turkey or whether the racial structure is preserved.

“The results of the study reported by Khaidizar et al. (2012) showed higher genetic polymorphism when they used SSR to investigate the level of polymorphism in Turkish dry bean genotypes, which includes most of the genotypes used in Ceylan et al.'s (2014) study. Consistent with several previous studies, cluster analysis revealed that it resulted in two major clusters, possibly representing two major gene pools, namely Andean and Mesoamerican. It was stated that these small-seeded cultivars, which clustered separately from the others in both plastid and nuclear marker analysis, may belong to the Mesoamerican gene pool.”

Ceylan, A., Öcal, N., Akbulut, M. (2014). Genetic diversity among the Turkish common bean cultivars (Phaseolus vulgaris L.) as assessed by SRAP, POGP and cpSSR markers. Biochemical Systematics and Ecology, 54, 219-229.

M.I. Khaidizar, K. Haliloglu, E. Elkoca, M. Aydın, F. Kantar Genetic dıversity of common bean (Phaseolus vulgaris l.) landraces grown in northeast Anatolia of Turkey assessed with simple sequence repeat markers Turk. J. Field Crops, 17 (2012), pp. 145-150

6.       L24: Why are SSRs the “best” markers for studying genetic diversity? SNPs are codominant, biallelic and much more abundant than SSRs.

Response:  You are right, it is a very ambitious sentence. SNP markers have been used instead of SSRs in some studies in recent years. However, SRRs give successful results in genetic diversity studies in beans. Based on your comment, the sentence has been edited as follows:

“Simple sequence repeats (SSRs), which are codominant markers, are preferred for the determina-tion of genetic diversity because they are highly polymorphic, multi-allelic, highly reproduci-ble, and have good genome coverage.”

7.       L49: Beans are not “the” primary source vegetable protein, but one of many. Change “the” to “a”.

Response:  Necessary adjustments have been made to the sentence.

Beans consumed in different forms (green pods, immature or dried seeds) are a primary source of vegetable protein

8.       L51: Is [3] the appropriate reference for this statement? It references a paper on genetic diversity (not nutrition) in Serbian beans.

Response:  Reference revised and new reference added.

In terms of genetic diversity, the Mesoamerican gene pool has more diversity than the Andean gene pool [Mamidi et al., 2013]

- Mamidi S, Rossi M, Moghaddam SM, Annam D, Lee R, Papa R, McClean PE: Demographic factors shaped diversity in the two gene pools of wild common bean Phaseolus vulgaris L. Heredity, 2013, 110(3), 267-276.

9.       L53: Use “Middle American” rather than “Central American” or “Mesoamerican” for center of domestication. Also, replace “gene centers” with the term “centers of domestication” which is much more descriptive. And to be clear, “Middle American” is used to describe the center of domestication to prevent confusion when discussing race “Mesoamerica”.

Response:  The word has been changed according to your suggestion.

Seeds of beans in the Middle American gene pool are characterized as either small or medium-sized,

10.    L54: Check that reference numbers are correct. [4] is about genetic diversity in Croatian beans but I think the information in this sentence is better supported by [5]. See also [7] which describes genetic diversity techniques and doesn’t mention beans at all.

Response: Necessary controls and arrangements were made.

These are Middle American gene centers. These centers of genetic diversity are separated from each other by both geographical and partial reproductive barriers [1, 5].

In terms of genetic diversity, the Mesoamerican gene pool has more diversity than the Andean gene pool [Mamidi et al., 2013]

11.    L59: Seeds of genotypes should be seeds of phenotypes

Response:  Necessary arrangements were made.

…… small or medium-sized, while seeds of phenotypes in the Andean gene pool…….

12.    L66-68: This statement about narrowing of the genetic base cites a paper from Brazil. Is there a more general reference?

Response: Necessary controls and arrangements were made.

However, the increase in the use of commercial varieties in recent years and the insufficient use of local varieties in breeding programs have led to a significant narrowing of the genetic base [Ewing et al., 2019]

- Ewing PM, Runck, BC, Kono TY, Kantar MB: The home field advantage of modern plant breeding. PloS one, 2019, 14(12), e0227079.

13.    L71: Rather than saying “the gene center for many plant species”, I would say “the center of domestication for important Old World cereal and grain legume crops”

Response:  The sentence has been corrected according to your suggestion.

In addition to being the center of domestication for important Old World cereal and grain legume crops with its ecological

14.     L95-96: I don't agree with this statement. The Illumina Beadchip is widely available & cost is ~$20/sample for the 12K chip, which is the cheapest of high-throughput technologies available. Its disadvantage is that you have to run all the SNPs, which is overkill if you are only interested in a specific chromosome region. GBS is more expensive but still under $50/sample for ~50K SNPs.

Response: 

Removed sentence….” The use of the 95 SNP chip method for beans is constrained by the high cost and the lack of available chips.”

15.    L123: Tables and figures in this paper generally do not have complete titles. Figures and tables should be able to be interpreted without reading the text, so table titles and figure legends should be detailed, and footnotes used to explain abbreviations etc. In the case of Table 1, its not clear what country/region these bean accessions come from and that they are being used for a genetic diversity study. Also, in table 1, its difficult to know which sets of accessions to which site numbers correspond. The site number should be in line with the first accession of its group.

Response:  Table and figure explanations have been edited considering your suggestion.

For example;

Table 1. List of bean inclusions according to information and coordinates of the collection location (Figure 1).

Table 2. SSR primers and sequence information used for genetic diversity analysis among bean accessions.

Table 3. Summary information obtained with twenty seven 27 SSR primer pairs used in bean accessions collected from İspir location

Table 4. Effective number of alleles (Ne), expected heterozygosity (He) and Shannon information index (I) based on 27 SSR loci bean accessions

In addition, the site numbers in Table 1 were tried to be made more understandable and the table was revised.

16.    L129: What are the modifications made to DNA extraction?

Response:  The relevant sentence is misspelled. The sentence was rearranged:

Türkiye. Genomic DNA extractions were performed as described by Zeinalzadehtabrizi et al. [29].

17.     L149: Need complete title for table 2. Also, there are some discrepancies in the table: Markers BM114 to BM175 belong to reference 34, not 32; Marker PV-AG004 belongs to reference 31, not 32. Did the forward and reverse sequences change in the PVBR14 marker? If so, what is the reason? Also, according to the cited source, the same marker was not used for the reverse primer (according to the reference DQ185880 was used) even though the forward primer belonged to the PVBR14 marker (there is a possibility that it will be PVBR20). If changes are to be made from the sources used, these changes should be stated. Has the reverse sequence of the BM153 marker been modified? The forward sequence of the GATS91 marker belongs to reference 34, not 32. The reverse sequence is not available in the source. PVTTTC001 should be PV-TTTC001 and belongs to reference 31, not 35. I could not find the first 4 markers in the given references.

Response:  The table title has been rearranged. In addition, the required reference controls and changes were made with the primer information.

-Updated reference for the first 4 primers.

-For first primer Update reference: Blair MW, Pedraza F, Buendia HF, Gaitán-Solís E, Beebe SE, Gepts P, Tohme J: Development of a genome-wide anchored microsatellite map for common bean (Phaseolus vulgaris L.). Theor. Appl. Genet. 2013, 107(8), 1362-1374.

-PVBR14 forward-backward sequence information did not change. However, we noticed that the sequence information was incorrectly written to the table. Necessary adjustments were made taking into account the reference of Buso et al. (2016).

New version: PVBR14:

ACGCTGTTGAAGGCTCTAC/ TGAGAAAGTTGATGGGATTG

-Based on your comment, we noticed that the reverse sequence of the BM153 pointer was misspelled. And after the literature check, we added the reverse sequence again.

BM153  Reverse(5’–3’):                TGACAAACCATGAATATGCTAAGA

18.    L180: Allele frequency ranged from 0.20 for BM153 to 0.78 for BM053 according to table 3 – rectify with the text.

Response:  Sentence corrected.

The allele frequency varied between 0.20 (BM153) and 0.78 (BM053).

19.    L200: Table 3 title needs to describe the crop, region and purpose of the study. Also, in Table 3, I am surprised to see such high levels of heterozygosity for some SSR markers since beans are highly self-pollinated.

Response:  The title of Table 3 has been rearranged based on your comment.

Table 3. Summary information obtained with twenty seven 27 SSR primer pairs used in bean accessions collected from İspir location

- Beans are a highly self-pollinating crop. However, the overall passthrough rate was relatively higher than predicted in the current study. This may be an indication that causes high genetic mixing in beans in İspir district or that beans come from different gene centers in the region.

20.    L231: Cluster A is cluster III in Fig 2 and group I in Table 5. Naming these three the same will clear up the confusion. In cluster A, it is said that there are 4 Ispir lines in the first subcluster. However, 5 types of Ispir lines are seen in Fig 2 and Table 5.

Response:  Considering your comment; Table 5 has been revised and aligned with figure 2. In Table 5, group 1 and group 3 were replaced. After the change, the spirit lines took their place in the appropriate cluster.

21.     L236-237: What appears to be gene flow among genotypes might be recent introduction from a common source.

Response:  Considering your comment; The relevant sentence has been revised as follows:

This result suggests that there may be some level of gene flow between genotypes or a recent introduction from a common source.

22.    L250-251: How did you determine genetic distance? There is no reference to a table or figure here and it is not possible to derive this information from Fig 2.

Response:  Calculation was made taking into account the genetic distance of Nei. However, results showing the distance between genotypes are not presented as a table. Genetic distance results were verbally stated in text only. The table showing the distance will be presented as a supplementary.

23.    L266: There are three samples (code #s 15, 16, and 17) taken from the Gaziler district. However, there are only two samples(15 and 16) in Fig 3. Was 17 not analyzed? Also, although sample 17 belongs to the Gaziler district, it appears to belong to the Yesilyurt village in Fig 3. I think there is slippage in the examples. If not, this difference should be explained.

Response: Thank you for your comment. As you mentioned, we noticed a shift in the samples. Figure 3 was rearranged by reanalyzing the data. And all the bean accessions took their place in the way

24.    Line 272: Sample 31, which belongs to the Ulubel village, is seen in the upper left side.

Response:  According to your comment, figure 3 has been reviewed and the sentence about Ulubel village has been revised.

“……….., while Ulubel Village genotypes are distributed over all parts of the diagram.”

25.    Line 273: YeÅŸilyurt village examples can be seen only in the lower left side. Fig 3, not 2.

Response:  Figure citation corrected according to your comment.

YeÅŸilyurt Village genotypes were located on lower left sections of Axis 1 (Figure 3).

26.    Line 274-275: Comparisons were made between samples representing only 3 regions. What is the reason of this?

Response:  Comparing 3 regions does not mean that the others are important or unimportant. It was stated that only the genetic diversity among these regions was less than the diversity among other regions.

27.     L283: Data in Table 6 is adequately described in the text and can be deleted.

Response: Table 6 has been deleted based on your suggestion. And the tables have been renumbered.

28.    L 284: Fig 3: Do the numbers given in the figure represent the code numbers of the samples given in Table 1? Necessary explanation should be given under the figure. Although a total of 45 accessions were used in the study, there are just 44 accessions in Fig 3.

Response:  Yes, these numbers represent the codes for participation. And this explanatory sentence was added as an explanation under the figure.

Figure 3. Principal coordinate analysis using SSR primer and separation on a two-dimensional diagram. The numbers in this figure represent the code numbers of the bean accessions presented in Table 1.

29.    L303: Text differs from table - The I value ranged from 0.055 (Ic) to 0.290 (mean 0.173) in Table 8.

Response:  Considering the relevant table; text edited.

The I value among the nine populations ranged from 0.055 (Ic) to 0.290 (Ov) (Mean 0.173).

30.    Line 306: PPL mean value is 33.41% in Table 8.

Response:  The value in the sentence has been corrected.

PPL value ranged from 9.15% (Ic) to 64.08% (Ov) (Mean 33.41%).

31.    Line 333: The expression states that there are 14 genotypes in the third subgroup. Specify if example 5 (Line-6) belongs to the third subgroup.

Response:  In Table 9, it was emphasized that Line-6 belonged to 3 subgroups.

32.    The figure on page 13 and line 327 should be Fig 4.

Response:  Necessary adjustment has been made.

33.    The figure on page 14 and line 335 should be Fig 5.

Response:  Necessary adjustment has been made.

34.     Figure 5 (former Fig 4) seems to be based on PCoA rather than structure analysis. Also, red and green colors in the pie charts in this figure are difficult for those with red-green color blindness to see. The red and green colors in the STRUCTURE analysis figure are ok.

Response:  Figure 5 was created based on structure analysis. in addition, the color tones in the pie chart colors in figure 5 were correlated with the structure analysis colors. if you have different color suggestions, we can rearrange them accordingly.

35.     L350: Table 10 – what do the bolded number mean?

Response:  Bold letters in the table are used to highlight which subgroup the participants belong to.

In addition, the "*" symbol has been added next to the bold letters and what the bold letters mean is added to the bottom of the table.

“*Bold letters in the table indicate which subgroup the participants belong to.”

Round 2

Reviewer 1 Report

Dear Authors,

Reviewer comments genes-1809872.R1

The revised manuscript entitled „Determining genetic diversity and population structure of common bean (Phaseolus vulgaris L.) landraces from Türkiye using SSR markers“ was modified by the authors in accordance with my previous comments.

I can now recommend the revised manuscript for publication in Genes.

I have only a few formal comments on the revised manuscript related to English language and style which are given below.

Formal comments:

Line 163: Add the word „respectively“ at the end of the statement „As a result of the test, the highest cophenetic coefficient was selecetd for genetic distance and clustering method Dice and UPGMA, r=0.745, respectively.“

Line 229: Add the word „respectively“ in the statement „Zargar et al. (45) determined the expected heterozygosity values as 0.2192 in the first subpopulation, 0.2124 in the second subpopulation, and 0.2821 in the third subpopulation, respectively, in their analysis…“

Line 249: Correct the typing error in the term „SSR analysis“.

Line 261: Add a comma following the word „however“ in the statement „However, Öztürk et al. (2)…“

Line 289: Add a comma between the words „together“ and „these three components…“, i.e., „…togetehr, these three components explained…“

Conclusions, line 393: Modify the statement as follows: „In the present study, genotypes collecteed from the Erzurum-Ispir region, located in the Northeastern Anatolia region of Türkyie, were evaluated at the molecular level…“

Final recommendation: Accept after a formal revision.

Author Response

 Responses to Comments of Reviewer 1

General Response:

Dear reviewer: We have made all necessary corrections and revised the manuscript based on your valuable suggestions and comments. We hope that revisions fulfill your expectations. New edits were highlighted in blue background color in the text.

Sincerely

Dr. Aras Turkoglu, Dr. Peter Poczai et al.

Comment

1.       Line 163: Add the word „respectively“ at the end of the statement „As a result of the test, the highest cophenetic coefficient was selecetd for genetic distance and clustering method Dice and UPGMA, r=0.745, respectively.“

Response:

Necessary correction was made.

Comment

2.       Line 229: Add the word „respectively“ in the statement „Zargar et al. (45) determined the expected heterozygosity values as 0.2192 in the first subpopulation, 0.2124 in the second subpopulation, and 0.2821 in the third subpopulation, respectively, in their analysis…“

Response: The sentence has been edited according to your suggestion.

Zargar et al. [45] determined the expected heterozygosity values as 0.2192 in the first subpopulation, 0.2124 in the second subpopulation, and 0.2821 in the third subpopulation, respectively, in their analysis using 15 RAPD and 23 SSR markers in 51 Indian bean genotypes.

Comment

3.       Line 249: Correct the typing error in the term „SSR analysis“.

Response: Necessary correction was made.

Comment

4.       Line 261: Add a comma following the word „however“ in the statement „However, Öztürk et al. (2)…“

Response: The word was corrected according to your suggestion.

However, Öztürk et al. [2], who…………

Comment

5.       Line 289: Add a comma between the words „together“ and „these three components…“, i.e., „…togetehr, these three components explained…“

Response: : The sentence was corrected according to your suggestion.

“……analysis was 20.57, 16.96, and 13.33; together, these three components explained 50.85% …….”

Comment

6.       Conclusions, line 393: Modify the statement as follows: „In the present study, genotypes collecteed from the Erzurum-Ispir region, located in the Northeastern Anatolia region of Türkyie, were evaluated at the molecular level…“

Response: : The sentence was changed.

“In the present study, genotypes collected from the Erzurum-Ispir region, located in the Northeastern Anatolia region of Türkiye, were evaluated at the molecular level.”

Reviewer 2 Report

The authors have improved the paper – particularly in terms of removing inconsistencies and errors in the text. A few remain and the authors did not respond directly to four points which I regard as serious flaws in the paper and need to be dealt with. These are:  

1) I am disappointed in the authors’ response that they cannot provide any phenotypic characteristics for associating genotypic groups with known biological groups. This would be easy to achieve even with limited germplasm and would bring the paper to a higher level of significance. At the very least, the authors should acquire seed weight even if only a few seeds are available (report as g/seed). Seed color would also be easy to obtain and with so few accessions, both traits could evaluated in an afternoon. At a bare minimum, the authors should include seed weight data for their accessions in table 1.

2)  The ultrameric property of UPGMA where it assumes that evolutionary change among lineages are all equal when in fact they rarely are. This is why Neighbor Joining (NJ) is often used where the more accurate but computationally intensive maximum likelihood or maximum parsimony methods are not possible. The authors should perform NJ and report the result.

3)      Related to point #2, the authors did not address my comment about finding the optimum tree. I suggested bootstrapping as a mean to establish confidence levels for nodes in the dendrogram and I strongly urge the authors to perform this analysis.

4)      The authors also dismissed my comment about whether any of the SSR markers were linked by saying that “…phenotypic features were not used in the study [and] mapping was not performed.”  I do not understand the logic here because phenotypic traits are not required to create a linkage map. In addition, I was not asking for the authors to construct a linkage map although in theory, this could be done using genome wide association mapping. I do agree that it would be difficult to construct a linkage map with the existing population and set of markers because of the limited number of accessions and sparse genome coverage with so few markers. However, the authors do not need to construct their own linkage map and can use data obtained the papers that originally described and mapped these markers. Below, I have provided below a list of the markers and the chromosome to which they are located. You can see that a number do fall on the same chromosome (especially Pv02, 04 07 and 09), and some may be linked while others may be distant enough to segregate independently. The authors should include chromosome location for the markers in their table on SSR characteristics. The authors should also determine whether any markers are closely linked. At a very minimum, they should report any linkages because one of the assumptions of a phylogenetics study is that markers are unlinked. The authors should consider thinning to a single marker in genomic regions where there is tight linkage and repeating the analysis. The author should also note that two chromosomes (Pv10 and Pv11) are apparently not covered by markers (although two SSRs are unlinked and might possibly fall on one or both of these chromosomes). Overall, it appears that some portions of the genome are over represented and others are not represented at all.  

Chromosome   SSr

Pv01:               BM200

Pv02:               Bmd-18, BM143, BM152, BM156, BM167, GATS91

Pv03:               Bmd-1

Pv04:               Bmd-15, BM161, PV-AT001, PV-AG004, BM199

Pv05:               Bmd-53, BM175

Pv06:               BM137

Pv07:               BM160, BM183, BM210, PV-TTTC001

Pv08:               BM211

Pv09:               BM114, BM141, BM154, BM188

Unlinked:        BM153, PVBR14 

Some line by line comments: 

L52: You say that the reference has been revised and new reference added but [3] is the only one and still about Serbian genetic diversity in beans characterized by SSRs, but the sentence in the present MS is about traditional uses of beans in Europe. I think a reference that directly supports your statement is required. 

L54: “and Andean” needs to be added after Middle American. 

L61-63: Information about the introduction of beans into Türkiye after the Columbian exchange began has been added but is somewhat confusing. It sounds like beans were introduced with the 1st Columbian expedition and then later in the 16th and 17th centuries. There is no hard evidence that beans were introduced with any of the Columbian expeditions or by other explorers until Pizzaro’s expedition along the Pacific coast of South America in 1528. Written records describing common bean in Europe date to 1532. Pizzaro’s putative introductions would have been from the Andean center of domestication. While Spanish explorers may have observed small seeded race Mesoamerica beans when they landed on the coast of Central America, they would not have observed medium seeded race Durango types (that are historically prevalent in Türkiye) until Cortez reached the arid highlands in the interior of Mexico in 1520. There is no evidence that his expeditions returned beans to Europe from Middle America. I think it’s best to just say: “It is thought that common bean was introduced to Europe in the 16th and 17th centuries, and it was introduced to Türkiye in the 17th century.” 

L72: A reference has been added, but this one appears to be general rather than specific to common bean. A better reference for [10, 11] would be Singh, S.P. (2001), Broadening the Genetic Base of Common Bean Cultivars. Crop Sci., 41: 1659-1675. https://doi.org/10.2135/cropsci2001.1659.  

Table 1: The heading for the second column should be Accession number.  

Table 2: Some marker names have a hyphen and others do not. Make sure you are consistent with the literature (BMd18 or BMd-18?). BM053 should be BMd-53. Forward and reverse primers are flipped for PVBR14. Check and edit the forward sequence of the GATS91 primer and both sequences of the PV-TTTC001 primer. The GATS91 primer is available in both [33 and 36]. However, the reverse sequence in both references is completely different. Correct the existing error. 

L174-175: There are two references in the reference list designated as [40]. Is Lewontin (1972) cited at all in the present MS? 

L185: Nm is only mentioned here and is undefined in the manuscript. Is it necessary to telling your story?  

L252: There are 5 Ispir lines in subcluster I, not 4. 

L268-269: This sentence does not provide an explanation as to why different markers would produce different numbers of subgroups. You need to dig deeper or say you don’t know. How do the 5 clusters in [57] relate to the 3 clusters in the present MS? Are some of the 5 subgroups of the 3? Are the same accessions in each study grouped together in corresponding clusters? This assumes that you can match the clusters by the accessions each contains.  

Table 5: Table title is uninformative – groups of what? What is the study? etc. 

In the authors’ response to #5, the authors have provided a nice explanation but it should be incorporated into the manuscript. The first reference [18] is mentioned at L94-95 and the second [56] at L258-260, but there needs to be a synthesis in the discussion and a cleaned up version of the paragraph from the response to reviewer would serve.

Author Response

Responses to Comments of Reviewer 2

General Response:

Dear reviewer: We have made all necessary corrections and revised the manuscript based on your valuable suggestions and comments. We hope that revisions fulfill your expectations. New edits were highlighted in blue background color in the text.

Sincerely

Dr. Aras Turkoglu, Dr. Peter Poczai et al.

1.       I am disappointed in the authors’ response that they cannot provide any phenotypic characteristics for associating genotypic groups with known biological groups. This would be easy to achieve even with limited germplasm and would bring the paper to a higher level of significance. At the very least, the authors should acquire seed weight even if only a few seeds are available (report as g/seed). Seed color would also be easy to obtain and with so few accessions, both traits could evaluated in an afternoon. At a bare minimum, the authors should include seed weight data for their accessions in table 1.

Response:

Some seed characteristics (100-seed weight, seed color and Seed shape) were added to Table 1, based on your recommendation.

2.       The ultrameric property of UPGMA where it assumes that evolutionary change among lineages are all equal when in fact they rarely are. This is why Neighbor Joining (NJ) is often used where the more accurate but computationally intensive maximum likelihood or maximum parsimony methods are not possible. The authors should perform NJ and report the result.

Response:

Considering your suggestion, the phylogenetic tree was reconstructed by the neighbor-joining method. Previousr dendrogram was removed. After this change, figure 2 and table 5 were rearranged according to the new dendrogram. Also, the method section has been updated.

“In this study, Phylogenetic analysis was performed with MEGA 6.0 software. The dendrogram was constructed using the neighbor-joining method of the MEGA software with the maximum composite likelihood substitution model, and bootstrapping with 1,000 replicates [37].”

Figure 2. Dendrogram showing the genetic relationship between 45 bean genotypes generated by the neighbor-joining method of the MEGA software with the maximum composite likelihood substitution model using 27 SSR markers.

Table 5. Groups and subgroups of Phaseolus vulgaris accessions determined as a result of Neighbor Joining (NJ) cluster analysis.

3.       The authors also dismissed my comment about whether any of the SSR markers were linked by saying that “…phenotypic features were not used in the study [and] mapping was not performed.”  I do not understand the logic here because phenotypic traits are not required to create a linkage map. In addition, I was not asking for the authors to construct a linkage map although in theory, this could be done using genome wide association mapping. I do agree that it would be difficult to construct a linkage map with the existing population and set of markers because of the limited number of accessions and sparse genome coverage with so few markers. However, the authors do not need to construct their own linkage map and can use data obtained the papers that originally described and mapped these markers. Below, I have provided below a list of the markers and the chromosome to which they are located. You can see that a number do fall on the same chromosome (especially Pv02, 04 07 and 09), and some may be linked while others may be distant enough to segregate independently. The authors should include chromosome location for the markers in their table on SSR characteristics. The authors should also determine whether any markers are closely linked. At a very minimum, they should report any linkages because one of the assumptions of a phylogenetics study is that markers are unlinked. The authors should consider thinning to a single marker in genomic regions where there is tight linkage and repeating the analysis. The author should also note that two chromosomes (Pv10 and Pv11) are apparently not covered by markers (although two SSRs are unlinked and might possibly fall on one or both of these chromosomes).

Response:

- Considering your suggestions; In Table 2, information such as chromosome information (Linkage group), motifs, and Genebank codes belonging to SSR markers were added.

In addition, new sentence about the chromosome location of the SSR markers was added in the Material section.

There are three markers on the Pv01 chromosome, five markers on the Pv02 chromosome, five markers on the Pv04 chromosome, two markers on the Pv06 chromosome, two markers on the Pv07 chromosome, three markers on the Pv08 chromosome, and five markers on the Pv09 chromosome. There was only one marker on the Pv01 (BM053 marker) and Pv05 (BM175 marker) chromosomes. In addition, none of the markers used in our study were markers located on the Pv10 and Pv11 chromosomes (Table 2).

4.       L52: You say that the reference has been revised and new reference added but [3] is the only one and still about Serbian genetic diversity in beans characterized by SSRs, but the sentence in the present MS is about traditional uses of beans in Europe. I think a reference that directly supports your statement is required.

Response:  Added a new reference pointing to the relevant sentence.

For many people living in European countries, Phaseolus vulgaris is a traditional dietary component [3; Rodríguez et al., 2022].

“Rodríguez L, Mendez D, Montecino H, Carrasco B, Arevalo B, Palomo I, Fuentes E: Role of Phaseolus vulgaris L. in the prevention of cardiovascular diseases—cardioprotective potential of bioactive compounds. Plants, 2022, 11(2), 186.”

5.       L54: “and Andean” needs to be added after Middle American.

Response:  Word added.

These are Middle American and Andean gene centers.

6.       L61-63: Information about the introduction of beans into Türkiye after the Columbian exchange began has been added but is somewhat confusing. It sounds like beans were introduced with the 1st Columbian expedition and then later in the 16th and 17th centuries. There is no hard evidence that beans were introduced with any of the Columbian expeditions or by other explorers until Pizzaro’s expedition along the Pacific coast of South America in 1528. Written records describing common bean in Europe date to 1532. Pizzaro’s putative introductions would have been from the Andean center of domestication. While Spanish explorers may have observed small seeded race Mesoamerica beans when they landed on the coast of Central America, they would not have observed medium seeded race Durango types (that are historically prevalent in Türkiye) until Cortez reached the arid highlands in the interior of Mexico in 1520. There is no evidence that his expeditions returned beans to Europe from Middle America. I think it’s best to just say: “It is thought that common bean was introduced to Europe in the 16th and 17th centuries, and it was introduced to Türkiye in the 17th century.”

Response:  The sentence has been rearranged according to your suggestion.

It is thought that common bean was introduced to Europe in the 16th and 17th centuries, and it was introduced to Türkiye in the 17th century [8].

7.       L72: A reference has been added, but this one appears to be general rather than specific to common bean. A better reference for [10, 11] would be Singh, S.P. (2001), Broadening the Genetic Base of Common Bean Cultivars. Crop Sci., 41: 1659-1675. https://doi.org/10.2135/cropsci2001.1659. 

Response:  The reference you suggested was added.

However, the increase in the use of commercial varieties in recent years and the insufficient use of local varieties in breeding programs have led to a significant narrowing of the genetic base [10, 11].

10.             Singh SP: Broadening the genetic base of common bean cultivars: a review. Crop sci. 2001, 41(6), 1659-1675.

8.       Table 1: The heading for the second column should be Accession number. 

Response:  Necessary corrections have been made to the title in the table.

9.       Table 2: Some marker names have a hyphen and others do not. Make sure you are consistent with the literature (BMd18 or BMd-18?). BM053 should be BMd-53. Forward and reverse primers are flipped for PVBR14. Check and edit the forward sequence of the GATS91 primer and both sequences of the PV-TTTC001 primer. The GATS91 primer is available in both [33 and 36]. However, the reverse sequence in both references is completely different. Correct the existing error.

Response: Fixed inconsistency in marker names throughout the manuscript. In addition, the sequence information of the markers was rechecked and corrected.

10.    L174-175: There are two references in the reference list designated as [40]. Is Lewontin (1972) cited at all in the present MS?

Response:  The reference order has been corrected in the manuscript and in the reference section.

11.    L185: Nm is only mentioned here and is undefined in the manuscript. Is it necessary to telling your story? 

Response: The sentence has been rearranged.

Fst measures the amount of genetic variance that can be explained by population structure based on Wright’s F-statistics.

12.    L252: There are 5 Ispir lines in subcluster I, not 4.

Response:  The sentence has been corrected according to your suggestion.

Cluster III consists of two subgroups; Aras-98, Elkoca-05, Göynük-98, Yakutiye-98 and KaracaÅŸehir-90 cultivars were included in the first subcluster, along with five Ispir bean lines, and 63, 64, 65, 69 accessions were included in the second subcluster. In addition, cluster I consisted of 2 subgroups; There were twenty-three participants in the first subgroup and 4 participants in the second subgroup. In the cluster II, there were 8 participants. (Figure 2 and Table 5).

13.     L268-269: This sentence does not provide an explanation as to why different markers would produce different numbers of subgroups. You need to dig deeper or say you don’t know. How do the 5 clusters in [57] relate to the 3 clusters in the present MS? Are some of the 5 subgroups of the 3? Are the same accessions in each study grouped together in corresponding clusters? This assumes that you can match the clusters by the accessions each contains. 

Response: 

Removed sentence: “This difference can be explained by the different markers used in the  studies.”

14.    Table 5: Table title is uninformative – groups of what? What is the study? etc.

Response:  The title of Table 5 has been edited according to your suggestion.

Table 5. Groups and subgroups of Phaseolus vulgaris accessions determined as a result of Neighbor Joining (NJ) cluster analysis.

15.    In the authors’ response to #5, the authors have provided a nice explanation but it should be incorporated into the manuscript.

Response:  Sentence manuscript included.

“The results of the study reported by Khaidizar et al. [56] showed higher genetic polymorphism when they used SSR to investigate the level of polymorphism in Turkish common bean genotypes, which includes most of the genotypes used in Ceylan et al. [18] study. Consistent with several previous studies, cluster analysis revealed that it resulted in two major clusters, possibly representing two major gene pools, namely Andean and Mesoamerican. It was stated that these small-seeded cultivars, which clustered separately from the others in both plastid and nuclear marker analysis, may belong to the Mesoamerican gene pool.”
